# A satellite chronology of plumes from the April 2021 eruption of La Soufrière, St Vincent

Isabelle A. Taylor[1], Roy G. Grainger[1], Andrew T. Prata[2,a], Simon R. Proud[3,4,b], Tamsin A. Mather[5], and David M. Pyle[5]

[1]COMET, Atmospheric, Oceanic and Planetary Physics, University of Oxford, Oxford, OX1 3PU, UK
[2]Atmospheric, Oceanic and Planetary Physics, University of Oxford, Oxford, OX1 3PU, UK
[3]NCEO, Atmospheric, Oceanic and Planetary Physics, University of Oxford, Oxford, OX1 3PU, UK.
[4]NCEO, RAL Space, STFC Rutherford Appleton Laboratory, Harwell, OX11, UK.
[5]COMET, Department of Earth Sciences, University of Oxford, Oxford, OX1 3AN, UK
[a]Now at: School of Earth, Atmosphere and Environment, Monash University, Clayton, Victoria 3800, Australia
[b]Now at: RAL Space, STFC Rutherford Appleton Laboratory, Harwell, OX11, UK.

**Correspondence:** Isabelle A. Taylor (isabelle.taylor@physics.ox.ac.uk)

**Abstract.** Satellite instruments play a valuable role in detecting, monitoring and characterising emissions of ash and gas into the atmosphere during volcanic eruptions. This study uses two satellite instruments, the Infrared Atmospheric Sounding Interferometer (IASI) and the Advanced Baseline Imager (ABI), to examine the plumes of ash and sulfur dioxide ($SO_2$) from the April 2021 eruption of La Soufrière, St Vincent. The frequent ABI data has been used to construct a 14-day chronology of a series of explosive events at La Soufrière; which is then complemented by measurements of $SO_2$ from IASI which is able to track the plume as it is transported around the globe. A minimum of 35 eruptive events were identified using true, false and brightness temperature difference maps produced with the ABI data. The high temporal resolution images were used to identify the approximate start and end times, as well as the duration and characteristics of each event. From this analysis, four distinct phases within the 14-day eruption have been defined, each consisting of multiple explosive events with similar characteristics: (1) an initial explosive event, (2) a sustained event lasting over nine hours, (3) a pulsatory phase with 25 explosive events in a 65.3 hour period and (4) a waning sequence of explosive events. It is likely that the multiple explosive events during the April 2021 eruption contributed to the highly complex plume structure which can be seen in the IASI measurements of the $SO_2$ column amounts and heights. The bulk of the $SO_2$ from the first three phases of the eruption was transported eastwards, which based on the wind direction at the volcano, implies the $SO_2$ was largely in the upper troposphere. Some of the $SO_2$ was carried to the south and west of the volcano, suggesting a smaller emission of the gas into the stratosphere: there being a shift in wind direction around the height of the tropopause. The retrieved $SO_2$ heights show that the plume had multiple layers but was largely concentrated between 13 and 19 km, with the majority of the $SO_2$ being located in the upper troposphere and around the height of the tropopause, with some emission into the stratosphere. An average e-folding time of $6.07 \pm 4.74$ days was computed based on the IASI $SO_2$ results: similar to other tropical eruptions of this magnitude and height. The $SO_2$ was trackable for several weeks after the eruption and is shown to have circulated the globe with parts of it reaching as far as $45°$ S and $45°$ N. Using the IASI $SO_2$ measurements, a timeseries of the total $SO_2$ mass loading was produced, with this peaking on 13 April (descending orbits) at $0.31 \pm 0.09$ Tg. Converting these mass values to a temporally varying $SO_2$ flux demonstrated

that the greatest emission occurred on 10 April with that measurement incorporating $SO_2$ from the second phase of the eruption (sustained emission), and the beginning of the pulsatory phase. The $SO_2$ flux is then shown to fall during the later stages of

the eruption: suggesting a reduction in eruptive energy, something also reflected in ash height estimates obtained with the ABI instrument. A total $SO_2$ emission of $0.63 \pm 0.5$ Tg of $SO_2$ has been derived, although due to limitations associated with the retrieval, particularly in the first few days after the eruption began, this, the retrieved column amounts, and the total $SO_2$ mass on each day should be considered minimum estimates. There are a number of similarities between the 1979 and 2021 eruptions at La Soufrière, with both eruptions consisting of a series of explosive events with varied heights and including some emission

into the stratosphere. These similarities highlight the importance of in-depth investigations into eruptions, and the valuable contribution of satellite data for this purpose, as such studies help to learn about a volcano's behaviour, which may help to be better prepared for future eruptive activity.

## 1  Introduction

La Soufrière ($61.18°$ W, $13.33°$ N, summit elevation of 1220 m), a volcano on the island of St Vincent, entered a phase of

explosive activity on 9 April 2021 after having been in a lower level state of eruption, including the slow extrusion of a lava dome, since late December 2020 (Global Volcanism Program, 2021a; Joseph et al., 2022). The volcano is part of the Eastern Caribbean volcanic arc and has erupted on at least five occasions since the 18[th] century in both explosive (1718, 1812, 1902-3, 1979) and lava dome-forming eruptions (1971-2, 1979) (Robertson, 1995; Pyle, 2017).

Eruptive activity since 1970 included the non-explosive extrusion of a basaltic andesite lava dome into the flooded crater

from November 1971 - January 1972; and a violent series of explosions that began in the flooded summit crater on 13 April 1979 (Aspinall et al., 1973; Shepherd et al., 1979). The 1979 explosions were very well documented at the time and persisted for two weeks. They were followed by about six months of lava dome extrusion across the crater floor, which by then had been infilled by pyroclastic ejecta (Brazier et al., 1982; Fiske and Sigurdsson, 1982; Shepherd and Sigurdsson, 1982). The explosive eruption of La Soufrière in April 1979 was one of the first eruptions to have occurred during the "satellite era" (Fiske

and Sigurdsson, 1982). Observations from the infra-red radiometer on the Synchronous Meteorological Satellite - 1 (SMS-1) were used to measure the growth of the volcanic ash plumes (Krueger, 1982); the Stratospheric Aerosol and Gas Experiment (SAGE) measured stratospheric aerosol from the eruption (McCormick et al., 1982); and $SO_2$ from the eruption was observed by the Total Ozone Mapping Spectrometer (TOMS; Carn et al. 2003, 2016).

After the 1979 eruption, La Soufrière showed no detectable signs of activity or unrest until late 2020. An increase in seis-

micity was noted by seismologists at The University of the West Indies Seismic Research Centre (UWI-SRC) in November and December 2020 (Joseph et al., 2022). An effusive eruption began on 27 December 2020 with the emplacement of a new lava dome, which grew over the following months. On 8 April 2021, the alert level was raised to the highest level and an evacuation of the highest risk communities was ordered (Global Volcanism Program, 2021b). The first of a number of explosive eruptive events began on 9 April 2021 at 08:41 LT (12:41 UTC) (Joseph et al., 2022). Multiple explosive events occurred over the

following two weeks, with the last explosive event occurring on 22 April 2021. Activity during this eruptive period led to the

closure of local airports and ash fall affected much of St Vincent, as well as neighbouring islands including the Grenadines, Barbados and Saint Lucia (Global Volcanism Program, 2021b). Details of the eruption, its impacts and the crisis management can be found within a special issue on the eruption published by the Geological Society (Robertson et al., 2024).

Technological developments in the four decades since the 1979 eruption mean that numerous aspects of the 2021 eruption were observed with multiple satellite instruments, including thermal anomalies, dome growth, lightning and the evolution of the $SO_2$, ash and sulfate plumes (Global Volcanism Program, 2021a, b; Smart and Sales, 2021; Babu et al., 2022; Thompson et al., 2022; Yue et al., 2022; Horváth et al., 2022; Dualeh et al., 2023; Bruckert et al., 2023; Camejo-Harry et al., 2024; Esse et al., 2024). Measurements made by satellite instruments allow the detection of $SO_2$ and ash, and quantification of the plume properties, which is essential for assessing the potential hazard to aviation (Prata and Tupper, 2009; Thomas and Watson, 2010; Lechner et al., 2017) and providing more sophisticated information on eruption source parameters (Aubry et al., 2021). Of particular value is the ability of satellite instruments to track the evolution of these plumes as they are transported away from the source. Geostationary instruments with a high temporal resolution (e.g. up to 30 seconds) are extremely valuable for identifying hazardous plumes and can also help to characterise eruptive events (e.g. Gupta et al., 2022; Prata et al., 2022).

This paper uses data from the Infrared Atmospheric Sounding Interferometer (IASI) on the three MetOp satellites, and the Advanced Baseline Imager (ABI) on the Geostationary Operational Environmental Satellite (GOES) to analyse the plumes of ash and $SO_2$ from the 2021 eruption of La Soufrière. Further details on the instruments and methods used to retrieve information about the plumes are given in section 2. The eruption sequence and plume characteristics are discussed in section 3. These instruments provide a complementary view of the eruption with the high temporal resolution of ABI allowing the eruption sequence to be evaluated, while using IASI it possible to study the dispersion of the plumes as they travel across the globe.

## 2 Methods

### 2.1 The Infrared Atmospheric Sounding Interferometer

#### 2.1.1 Instrument

The Infrared Atmospheric Sounding Interferometer (IASI) is a Fourier transform spectrometer on-board three meteorological satellite instruments: MetOp-A, -B and -C launched in 2006, 2012 and 2018 respectively, with data from all three instruments used in this study. The instrument's field-of-view consists of four circular pixels, each with a 12 km diameter (at nadir) within a 50 by 50 km square at nadir (Clerbaux et al., 2009). The instruments measure across a wide spectral range within the infrared part of the electromagnetic spectrum between 645 and 2760 $cm^{-1}$ (3.6 - 15.5 $\mu$m) with a high spectral sampling of 0.25 $cm^{-1}$ and apodized spectral resolution of 0.5 $cm^{-1}$ (Blumstein et al., 2004). More information on the IASI instrument can be found in Clerbaux et al. (2009). Within the instrument's spectral range there is sensitivity to volcanic ash (v-shaped absorption feature between 750 and 1250 $cm^{-1}$; Clarisse et al. 2010a) and $SO_2$ (three absorption features $\nu_1$, $\nu_3$ and $\nu_1+\nu_3$, centred at 8.7, 7.3 and 4 $\mu$m respectively). A number of methods have been developed to extract information about $SO_2$ from the IASI spectra (e.g.

Clarisse et al., 2008; Walker et al., 2011; Clarisse et al., 2012; Carboni et al., 2012; Walker et al., 2012; Clarisse et al., 2014). Retrieval techniques have also been developed for other volcanic gases and aerosols including $H_2S$ (Clarisse et al., 2011), CO (Martínez-Alonso et al., 2012), sulfate (Guermazi et al., 2021) and ash particles (Clarisse et al., 2010a, b; Maes et al., 2016; Ventress et al., 2016; Taylor et al., 2019). Each IASI instrument obtains near-global coverage twice a day, and being infrared measurements means that there is no break in coverage at night and during high latitude winters. This coverage and sensitivity to gases and aerosols associated with volcanic eruptions makes the IASI instruments well-suited for studying the evolution of volcanic plumes. This study uses retrieval schemes developed by Walker et al. (2011, 2012) and Carboni et al. (2012) for quantifying $SO_2$ emissions from La Soufrière and Sears et al. (2013) for the detection of volcanic ash.

### 2.1.2 Retrievals

In this study, two methods have been used to analyse $SO_2$ plumes from the La Soufrière eruption. The first method is a linear retrieval which is applied in this case to detect pixels which contain elevated quantities of columnar $SO_2$, see Walker et al. (2011, 2012). The second method is an optimal estimation retrieval scheme, which has been applied to the flagged pixels to quantify the column amount, height and the effective radiating temperature, and the errors associated with each of these. In broad terms, this method works by comparing the IASI measured spectra against $SO_2$ spectra simulated by the fast-radiative transfer model RTTOV (version 9; Saunders et al. 1999), see Carboni et al. (2012, 2016, 2019) for more details. A few changes have been made to the retrieval setup:

– A higher *a priori* height has been used. In most previous applications, an *a priori* height of 400 hPa ($\sim$7.6 km) has been used. This has been changed to 150 hPa ($\sim$14.3 km; upper troposphere) to reflect the higher injection height of the La Soufrière eruption as reported in Global Volcanism Program (2021b). The *a priori* variance has been kept at 500 hPa.

– In previous applications of the retrieval, the $1\sigma$ plume thickness was constrained and set to 100 hPa. This is more appropriate for a plume in the lower troposphere ($\sim$1.9 km at 400 hPa), but due to the exponential change in pressure with height in the atmosphere, this thickness is inappropriate in the upper troposphere and lower-stratosphere (UTLS) ($\sim$4.8 km at 150 hPa). In this application the *a priori* plume thickness has been set to 30 hPa ($\sim$1.2 km at 150 hPa): this is the minimum thickness that can be set given the spacing of the pressure levels in the RTTOV forward model.

The results have been divided into descending (satellite travelling N to S; $\sim$ 9:30 a.m. local overpass time at the equator) and ascending (satellite travelling S to N; $\sim$ 9:30 p.m. local overpass time at the equator). This approach can lead to artefacts in the results at the point where the data crosses into the next day. This is particularly notable in the ascending orbits at La Soufrière, with the date change occurring around the location of the volcano. To minimise this impact, the descending and ascending results have been offset from each other: the descending results are a 24 hour composite of the descending nodes of orbits starting on each date, while the ascending results are compiled from 24 hours of ascending nodes of orbits starting from midday on each date and up until midday of the following day. In this way the artefact is moved to the other side of the globe, so minimising its impact. In this paper, the ascending results are referred to by the start date.

**Table 1.** Percentage of IASI SO$_2$ pixels on 9 to 12 April 2021 which may be affected by volcanic ash. This has been calculated for the region -5 to 25° N, -68 to -20° E. AOD refers to ash optical depth. Note that the results are split into descending and ascending nodes, with the ascending results consisting of orbits beginning after 12:00 UTC on the date indicated, and ending at 12:00 UTC of the following day. See main text for further information.

| Date (dd/mm/yyyy) and orbit direction | Number of pixels containing SO$_2$ | % SO$_2$ flagged pixels containing ash with AOD $\geq 1$ | % SO$_2$ flagged pixels containing ash with AOD $\geq 2$ | % SO$_2$ flagged pixels containing ash with AOD $\geq 5$ |
|---|---|---|---|---|
| 09/04/2021 Ascending | 560 | 18.57 | 14.82 | 11.61 |
| 10/04/2021 Descending | 3645 | 15.8 | 11.63 | 7.71 |
| 10/04/2021 Ascending | 6865 | 9.45 | 6.57 | 3.57 |
| 11/04/2021 Descending | 12408 | 3.8 | 2.07 | 1.22 |
| 11/04/2021 Ascending | 16102 | 0.42 | 0.17 | 0.05 |
| 12/04/2021 Descending | 16932 | 0.12 | 0.06 | 0.03 |

To estimate the total mass of SO$_2$ the retrieved column amounts are first gridded to a regular grid of 0.125 by 0.125°(roughly 13.5 by 13.91 km at 13° N, 61° W): the retrieved values are interpolated to fill gaps created by the IASI field-of-view and where pixels fail the quality control. Using the area of each grid box, the column amounts are converted to a mass and the total mass on each day is then computed by summing all the gridded masses within a defined region (-45 to 45° N and -180 and 180° E). The total mass errors are computed in the same way which may lead to an overestimation of the mass error, but to sum the errors in quadrature could lead to an underestimation due to the systematic errors. Note that this overestimation in the errors is carried forward into the flux and total emission errors. A vertical distribution of the SO$_2$ mass can be obtained by combining the mass grid with a height grid generated from the retrieved heights.

Error analysis by Carboni et al. (2012) explored the effects of cloud and volcanic ash on the iterative SO$_2$ retrieval, demonstrating that ash with an optical depth of 1 (at 550 nm) can significantly affect the output of the retrieval. An ash optical depth of 2 was shown to cause a 50 % underestimation in the SO$_2$ column amount, and an ash optical depth of 5 was shown to mask the SO$_2$ signal completely. To investigate the effect of ash on the La Soufrière results, a linear ash retrieval as described in Sears et al. (2013) has been applied to the IASI data, for the first few days after the first eruptive event, to detect pixels containing volcanic ash which may affect the retrieved SO$_2$ values. This retrieval is run at three pressure levels (400, 600 and 800 hPa) to obtain three estimates of the ash optical depth. Pixels are then flagged as containing volcanic ash if any one of these ash optical depths exceeds a threshold. In this study, the results for the 400 hPa ($\sim$7.6 km) level are used as it is the closest level to the retrieved heights. The linear ash retrieval makes a number of assumptions which means that the retrieved ash optical depths are not necessarily accurate and it is being used here to simply indicate the possibility of ash affecting the SO$_2$ retrieval output. Note that there are spectral similarities between volcanic ash, desert dust and desert surfaces which can lead to false ash flags in desert regions such as the Sahara (e.g. Prata et al., 2001; Simpson et al., 2003; Park et al., 2014). To avoid this, the ash linear retrieval has been run for a smaller region (-5 to 25° N, -68 to -20° E) than the main analysis.

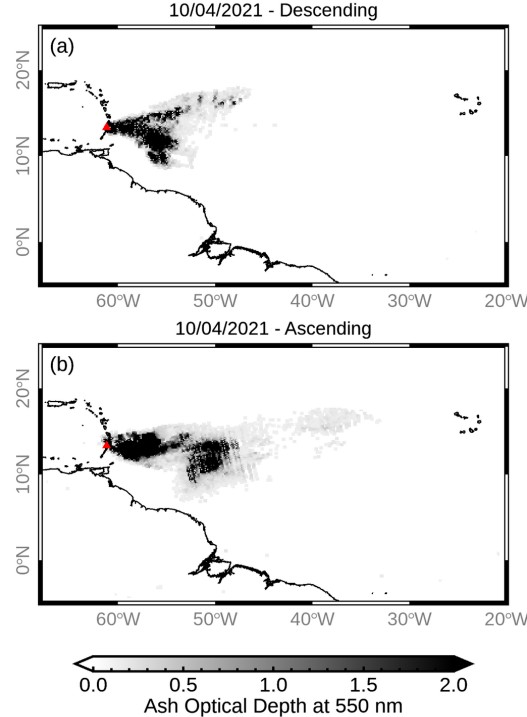

**Figure 1.** Ash optical depths at 550 nm from the IASI linear ash retrieval for 10 April 2021. This retrieval was run assuming a height of 400 hPa. The volcano's location is indicated by a red triangle. The results are a composite of multiple orbits which have been divided into descending ($\sim$ 9:30 a.m. local overpass time at the equator) and ascending ($\sim$ 9:30 p.m. local overpass time at the equator). The ascending results are a composite of orbits from 12:00 UTC on the 10 April to 12:00 UTC on 11 April. For further details see the main text.

Figure 1 shows examples of an optically thick ash plume travelling east from the volcano on 10 April 2021. Analysis of the percentage ash-detections (table 1), show that the $SO_2$ column amounts may be strongly effected by the presence of ash, and so the column amount and mass values presented in this paper should be considered minimum estimates, especially in the first few days after the eruption began (before the ash falls out). Future work should consider the simultaneous retrieval of ash and $SO_2$ so as to reduce the impact of ash on the $SO_2$ iterative retrieval. Additionally, $SO_2$ may be missed where it falls below the retrieval detection limit.

### 2.1.3 Estimating $SO_2$ e-folding time and flux

The decrease in the total mass of $SO_2$ ($m$) in the atmosphere with time ($t$) can be described as:

$$\frac{dm}{dt} = -\frac{1}{\tau}m \tag{1}$$

Where $\tau$ is the average $SO_2$ e-folding time. The e-folding time refers to the lifetime of $SO_2$ in the atmosphere and incorporates the loss of $SO_2$ due to oxidation and deposition, and where the $SO_2$ amount falls below the detection limit of the instrument being used. Fitting Eq. 1 to the total $SO_2$ atmospheric burden obtained with a satellite instrument can give a simple estimate of the average e-folding time over a given period. For an eruption like La Soufrière, where there are multiple emission events, the total $SO_2$ mass loading is a function of both the average e-folding time and variable $SO_2$ flux ($f$):

$$m_i = m_{i-1}e^{-\frac{1}{\tau}\Delta t} + f\tau(1 - e^{-\frac{1}{\tau}\Delta t}) \tag{2}$$

Where $i$ is the time step and $\Delta t$ is the time interval between measurements.

Carboni et al. (2019) uses Eq. 2 within an optimal estimation approach to estimate both the $SO_2$ flux at each time step and an average e-folding time for the entire eruptive period. This approach has been applied here to the total masses obtained for La Soufrière. The average e-folding estimate can be strongly influenced by the *a priori* value, and so for this study, an independent
estimate of the e-folding time was computed by fitting Eq. 1 to the IASI $SO_2$ masses computed for the 23 to 30 April 2021 (the period after the last explosive event occurred). This is shown in Fig. 2 and produced an e-folding estimate of 5.47 days. This value was used as the *a priori* in the optimal estimation approach, to generate a time-varying emission flux for the 14-day emission period (9 to 22 April 2021). The optimal estimation scheme produced an average e-folding time of $6.07 \pm 4.74$ days. On days with no explosive events, as identified with the ABI data (see section 2.2.2), the *a priori* $SO_2$ flux and uncertainty are
set to 0.

To compute the total mass of $SO_2$ erupted over the studied period, the fluxes were summed and then multiplied by the time difference ($\Delta$t) between the descending and ascending orbits (assumed to be 12 hours). To estimate the total erupted mass error, the individual flux errors are summed in quadrature (i.e. a quadratic mean of the errors), and then multiplied by $\Delta$t. An upper and lower bound of the total emission estimate is obtained by summing the $SO_2$ flux plus/minus the errors at each time
step (excluding negative values), and multiplying by $\Delta$t.

## 2.2 The Advanced Baseline Imager (ABI)

### 2.2.1 Instrument

The Advanced Baseline Imager (ABI) is on-board three of the Geostationary Operational Environmental Satellite platforms: GOES-16 (or GOES-East), GOES-17 (or GOES-West) and GOES-18, launched in 2016, 2018 and 2022 respectively. The
GOES-16 instrument is used in this study. Recent images produced with this instrument can be accessed at https://www.star. nesdis.noaa.gov/goes/fulldisk.php?sat=G16 (webpage last accessed on 05/04/2023). The area imaged by GOES-16 (positioned at 75.2° W) covers most of the continental United States, Eastern Canada, Central and South America and the Caribbean, extending across the Eastern Pacific and a large part of the North Atlantic Ocean. The ABI instrument has sixteen channels spanning the 0.45 to 13.7 $\mu$m spectral range, with the spatial resolution for the different channels varying from 0.5 to 2 km at
nadir (Schmit et al., 2005). Having seven channels between 7.3 and 13.3 $\mu$m means that the instrument has sensitivity to ash

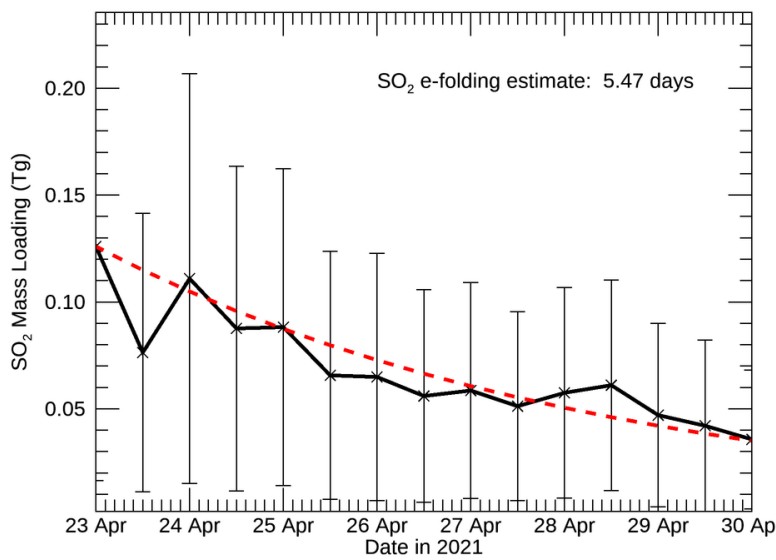

**Figure 2.** IASI SO$_2$ masses for 23-30 April 2021: the period after the last eruptive event at La Soufrière. A simple and independent estimate of SO$_2$ e-folding time has been computed by fitting Eq. 1 to the IASI total SO$_2$ mass estimates. This has been used as an independent *a priori* in the optimal estimation method for obtaining the flux and average e-folding time for the eruptive period (9-22 April).

and SO$_2$, and the instrument has previously been used for the detection of ash and SO$_2$, and quantification of ash properties (e.g. Pavolonis et al., 2020). Previous versions of the ABI instruments have also been used for this purpose (e.g. Yu et al., 2002; Ackerman et al., 2008). During the eruption of La Soufrière, the ABI sensor recorded a new full disc image every 10 minutes. In addition, on GOES-16 there are two moveable mesoscale regions, covering an area of 1000 x 1000 km (at subsatellite point),

which can provide data every minute (Schmit et al., 2017). These are moved to provide higher temporal coverage for events such as severe weather, hurricanes and forest fires (Schmit et al., 2017). During the La Soufrière eruption, one mesoscale region was moved over the volcano. This started at 9:00 UTC on 10 April (missing the earlier eruptive events) and ended at 06:00 UTC on 16 April 2021 (before the last three eruptive events). In this study, the ABI data has been used to identify the start and end times of each eruptive event, and for determining the height of the ash cloud. When available the 1-minute mesoscale data

have been used to identify the start and end times of the eruptive events. The 10-minute resolution data have been used for the remaining days. The 10-minute data has been used for the height analysis. While the ABI instrument has sensitivity to ash and SO$_2$, it has not been used to quantify the amount of either in this study. Instead, the ABI data has been used to document the sequence of explosive events produced by the 2021 La Soufrière eruption.

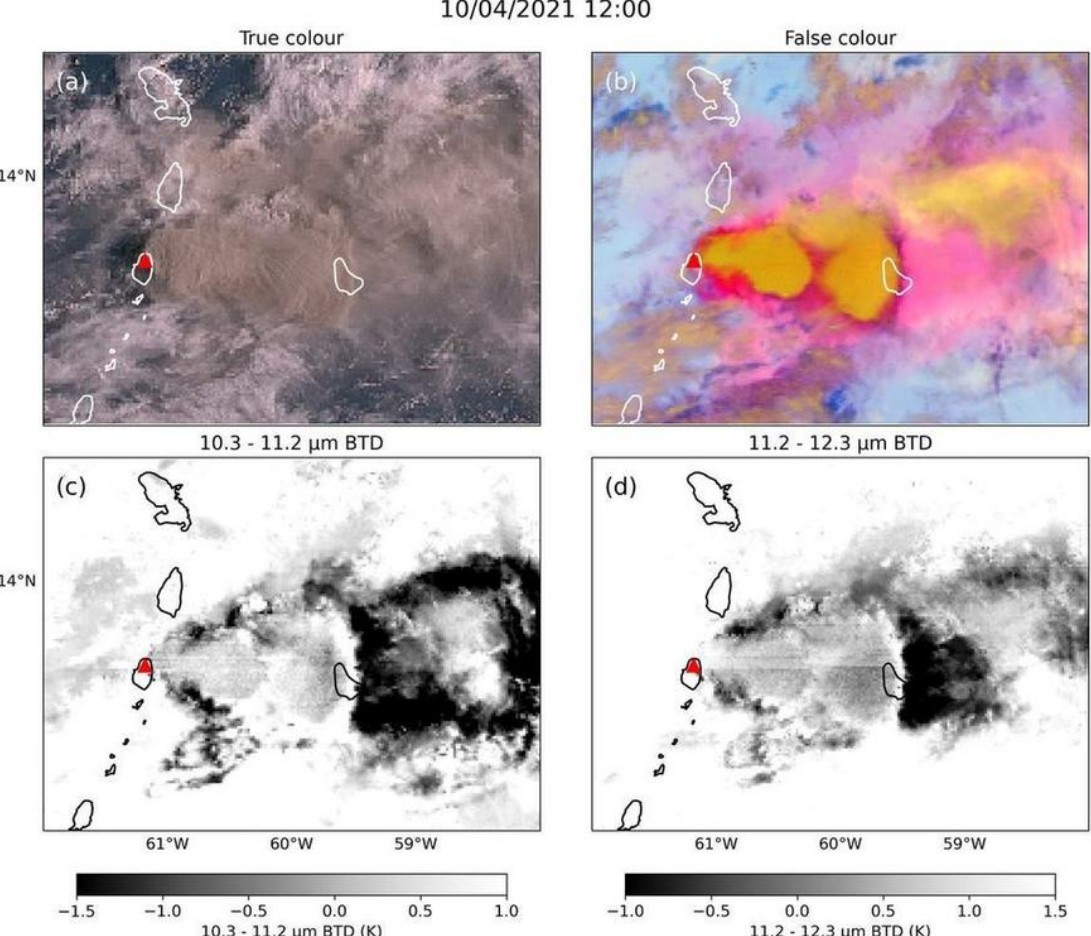

**Figure 3.** Examples of results from the ABI instrument for the La Soufrière plume at 12:00 on 10 April 2021. (a) True colour image. (b) False colour image (12.3 - 10.3 $\mu$m, 11.2 - 8.4 $\mu$m and 10.3 $\mu$m channels assigned to red, green and blue respectively). (c) 10.3 - 11.2 $\mu$m BTD. (d) 11.2 - 12.3 $\mu$m BTD. Plots for the period studied can be found in Taylor et al. (2023).

### 2.2.2 Analysis of ash

Utilising the Satpy Python package, true and false colour maps have been produced from the ABI data for the eruption period (9-22 April 2021). The false colour images have been constructed by assigning the 12.3 - 10.3 $\mu$m, 11.2 - 8.4 $\mu$m and 10.3 $\mu$m channels to red, green and blue respectively (for further information see GOES-R, 2018). Table 2 shows how to interpret the images. An example of these plots can be seen in Fig. 3. In addition, the 11.2 - 12.3 $\mu$m and 10.3 - 11.2 $\mu$m Brightness Temperature Differences (BTDs) have been calculated. The 11.2 - 12.3 $\mu$m BTD combination has been widely used to distinguish volcanic ash in satellite data and uses the positive transmission gradient between 10 and 12 $\mu$m caused by volcanic ash (Prata, 1989a, b). The 10.3 $\mu$m channel is less affected by water vapour (Lindsey et al., 2012) and so the 10.3 - 11.2 $\mu$m BTD may

**Table 2.** Outline of how to interpret the ABI false colour images (see GOES-R 2018 for more information). An example image is shown in Fig. 3.

| Colours | Explanation |
|---|---|
| Brown/Orange | Optically thick cloud (ice, ash or meteorological cloud) |
| Red/Pink | Less optically thick volcanic ash |
| Bright green/blue | $SO_2$ |

be more appropriate in tropical atmospheres. A threshold can be used to flag pixels containing ash. The BTD approach can be limited for a number of reasons, for example, false detections can occur in regions with high surface emissivities (e.g. deserts) and strong temperature inversions, or due to desert dust, while ash clouds may not be identified if they are too optically thin

or thick, or if there are significant quantities of ice or water within the plume (Rose et al., 1995; Simpson et al., 2000; Prata et al., 2001). Also there is no generally applicable threshold which can be set for ash detection with the most appropriate value varying with region and time.

Together the true, false and BTD images have been used to study the evolution of the plumes from La Soufrière. Careful examination of the plots allowed the identification of the approximate start and end times of each eruptive event. It should

be noted that the start and end times reported here will be different to those observed on the ground. Here the start time is the first time that the plume is observed in the ABI data, while the end time incorporates the time taken for the plume to rise and disperse away from the volcano. For this reason, the times reported here are different to those reported in Sparks et al. (2024) using seismic data. Additionally, the ABI analysis is limited by the 10/1-minute resolution of the instrument and can be complicated by the presence of cloud. The interpretation of the start and end times is a subjective process, with

the end time being particularly challenging to establish. In general, the end time was determined when the plume moved away from the volcano. This can be affected by the wind speed and so the end times should be considered approximate. As determining the end time was especially challenging an end time range has been given for most events. It is also likely that lower-magnitude explosive events or degassing between events may not be identifiable in the ABI data. Note that, for the full disc, the measurement time over La Soufrière has been computed based on the latitude of the volcano and the measurement

start and end times, approximately 243 seconds from the start time. This has been used to report the approximate times for the full disc results in this study. No such adjustment has been made to the mesoscale results, where the 1-minute temporal resolution ensures a higher accuracy.

An estimate of the ash height is obtained by comparing the brightness temperature in the 11.2 $\mu$m band (ABI channel 14) for each eruptive event in a 0.1° box around the volcano with the ERA-5 ECMWF temperature profile interpolated to the

volcano's location (European Centre for Medium-Range Weather Forecasts, 2021). Here this is referred to as the "Brightness Temperature (BT) method". The measured brightness temperature is representative of the temperature at the top of the ash or cloud layer and so can be used to approximate the height. The 11.2 $\mu$m channel is within an atmospheric window that has little sensitivity to water vapour. The minimum (or coldest) value within the 0.1° box around the volcano helps to select the optically thickest part of the plume and therefore the pixel which is least likely to be affected by radiation from beneath

the plume. The 0.1° box size should remove any significant effect due to parallax. In its application here, only one value is returned for each eruptive event (referred to as the "optically thick height method/solution") which ensures the result from the most optically thick part of the plume is reported but subsequently does not show any variability within the event. This method has been widely used to estimate the ash cloud top height (e.g. Prata and Grant, 2001). There are however a few limitations which are discussed in several papers (e.g. Oppenheimer, 1998; Prata and Grant, 2001; Zakšek et al., 2013), and which are

highlighted in Table 3. Despite the various limitations of this method, it is a quick and frequently used approach for estimating ash cloud heights. For this application an uncertainty of 1 K has been assumed for the instrument measurement encompassing the instrument noise and gaseous absorption above the cloud. Horváth et al. (2022) also applied the BT method to ABI data for La Soufrière, for comparison against GOES-17 side view heights, and noted a number of limitations with the BT method, including an underestimation of the heights for smaller eruptions due to a warm bias. The results here will likely share these

limitations and so should be treated with caution. However, Horváth et al. (2022) noted that the sideview and BT method heights showed better agreement for cold plumes or for plumes spreading around the height of the tropopause.

    In this study, there was commonly a tropospheric and stratospheric solution when using the BT method. Additional information has been obtained by comparing the plume direction and speed with the ERA-5 ECMWF wind profiles. Figure 4a shows the average and standard deviation of the wind direction with height and the average wind speed with height is shown in Fig.

4b. At heights of less than 5 km the wind is primarily travelling to the west (e.g. 267.5° at 4.3 km) with wind speeds of less than 10 m s$^{-1}$. At around 5 km the wind direction shifts to the east. Between 7 and 17 km the wind direction is fairly consistent (varying between 101 and 126°) before shifting back to the west at 19 to 20 km. The easterly winds in the stratosphere are a characteristic associated with a phase of the Quasi-Biennial Oscillation (QBO): alternating strong easterly and westerly zonal winds around the equator which propagate through the stratosphere to the tropopause (Reed et al., 1961; Baldwin et al., 2001).

The wind speed is shown to increase between 5 and 14 km and then fall between 15 and 20 km. Figure 4c and d show the average plume direction and speed. This has been computed by visually identifying the centre of the optically thick plume in the RGB image, an hour, or 30 minutes, after the start of each eruptive event (as identified with ABI). For the first 6 days of the eruption, where there is some uncertainty with regard to the height, the plume is shown to broadly be travelling to the east: supporting a tropospheric solution. To take this analysis further, at the location identified as the centre of the optically thick

plume, the BT method was used to obtain a second set of height solutions (referred to as the "BT method-centre" solutions). The wind directions at the heights obtained with both BT methods are compared against the horizontal velocity components from the motion of the plume (based on distance travelled in 1 hour or 30 minutes, as above) to assign a degree of confidence to the height results for each eruptive event. Figure 5 shows examples from the first three eruptive events which were shown to be tropospheric, uncertain and stratospheric respectively.

There are differences between the heights obtained with the "BT method" applied in this study and the results obtained using the BT method found in Horváth et al. (2022), which may arise from different sampling approaches. This is also the case for the BT heights presented in Sparks et al. (2024) where differences may also arise due to the different meteorological data and channels used.

**Table 3.** Summary of the sources of error associated with estimating the height of volcanic plumes by comparing the brightness temperature at 11.2 $\mu$m (ABI channel 14) with a temperature profile. In this paper this is termed the "Brightness Temperature (BT) method". These limitations are discussed in several papers (Oppenheimer, 1998; Prata and Grant, 2001; Zakšek et al., 2013).

| Limitation | Explanation | Mitigation (if applied) |
|---|---|---|
| **Single height estimate** | In this application this method only returns a single height rather than reflecting the heights across the plume. In addition, the minimum BT has been selected for each eruptive event and so the reported values do not reflect the variation in height during the eruptive event. | |
| **Temperature profile** | A good quality temperature profile, close to the volcano, is required (Oppenheimer, 1998; Zakšek et al., 2013) | An ECMWF ERA5 temperature profile is interpolated to the volcano's location. |
| **Optically thin ash** | When the plume is optically thin, upwelling radiation from beneath or within the plume contributes to the measured brightness temperature (Glaze et al., 1989; Zakšek et al., 2013) | This effect is minimised here by selecting the pixel with the coldest brightness temperature. This issue is may affect the later explosive events which have optically thinner ash. |
| **Poor correspondence between temperature profile and satellite measurement** | The method assumes that the ash is at an equal temperature to the surrounding air, if this is not the case, then the method will not perform well. Oppenheimer (1998) notes that the plume may not have reached its maximum height or that the momentum of the plume means it may have overshot the thermally equilibrium level. Some eruptions (e.g. El Chichón and Mt. St. Helens) the ash plume is much colder than the surrounding air and so the heights obtained were incorrect (Woods and Self, 1992; Holasek et al., 1996a). | If there is no intersection with the temperature profile no result is reported. |
| **Multiple solutions** | Multiple solutions can arise due to multiple intersections with the temperature profiles (e.g. above and below the tropopause, and other temperature inversions), in such cases, additional data sources are required to clarify the result (Oppenheimer, 1998) | In this study multiple solutions are reported. Further analysis using the plume's direction of travel and the wind profiles have been used to try and determine which solution is more appropriate (see main text). |
| **Isothermal atmospheres** | The method is limited in parts of the atmosphere with little temperature variation (Holasek and Self, 1995). A similar effect is observed using other infrared retrieval techniques (e.g. Prata et al., 2022). | |
| **Above cloud absorption** | This method does not account for any gaseous absorption above the cloud top | This is included within the 1 K uncertainty |
| **Meteorological cloud** | Where there is meteorological cloud overlying the plume, heights will reflect the height of the meteorological cloud layer. | A single height is reported for each event, typically near the start of the event when the ash is not obscured by cloud. |

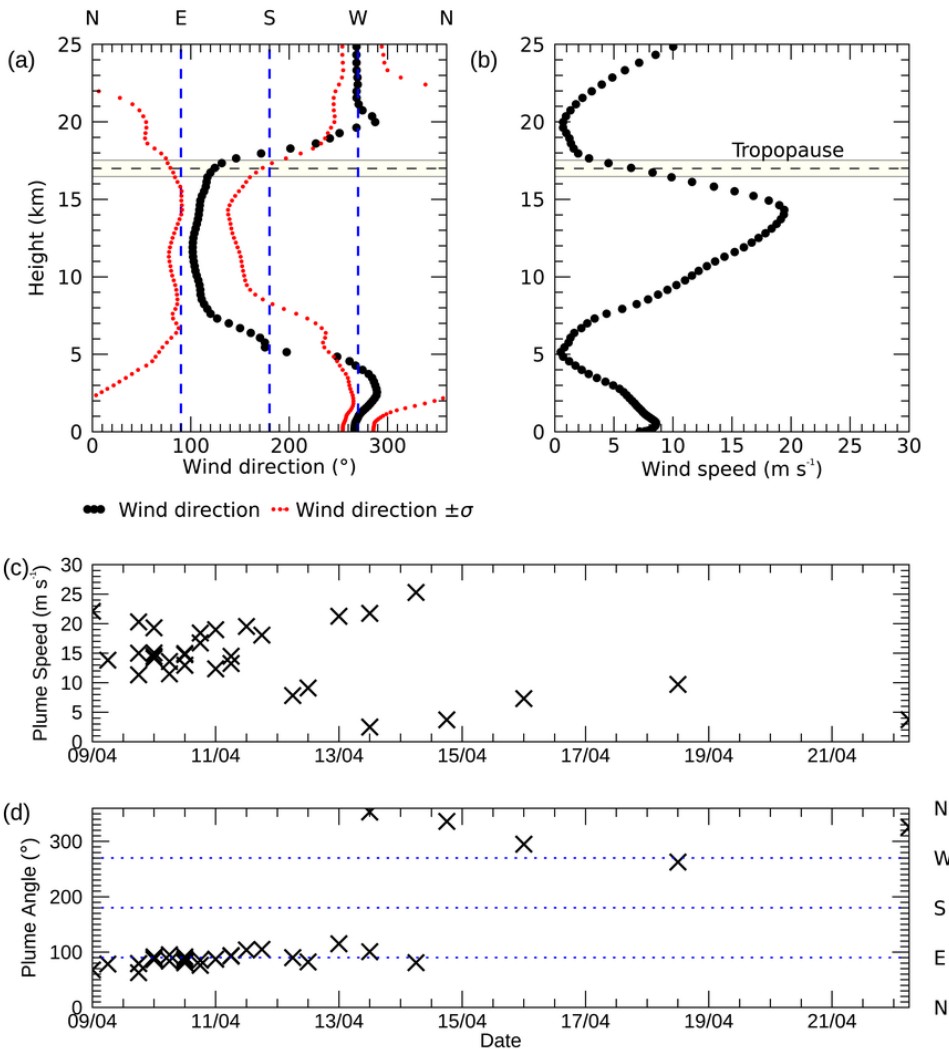

**Figure 4.** (a) The average and standard deviation of the ERA5 ECMWF wind direction for different height layers between 9 and 22 April. (b) Average wind direction profile of the ERA5 ECMWF wind direction for different height layers between 9 and 22 April. (c) Plume speed for the different explosive events. (d) Plume direction of travel for the different explosive events.

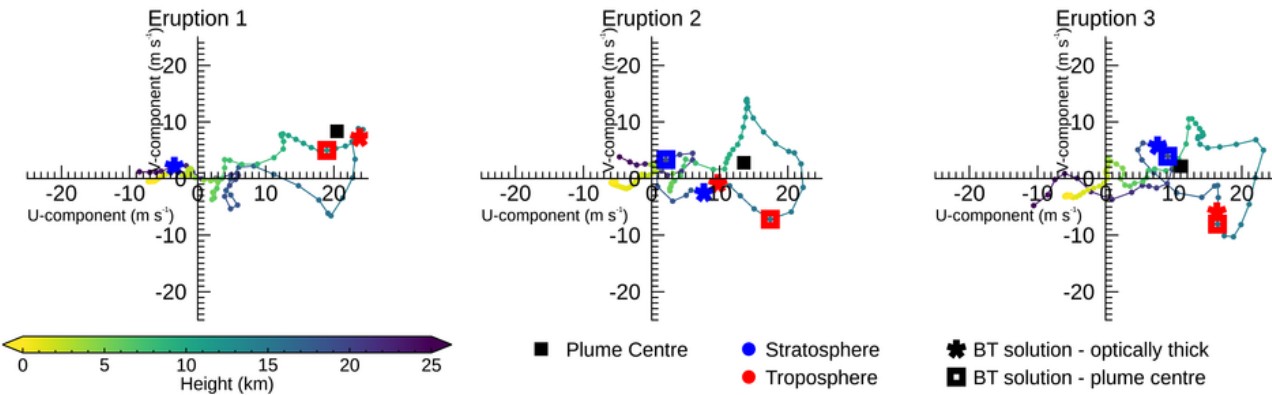

**Figure 5.** Demonstration of the use of the ERA5 ECMWF wind data to determine whether the tropospheric or stratospheric solution is more likely. Examples are from the first three eruptive events (see Table 4). These were classified as tropospheric, inconclusive and stratospheric respectively.

## 3 Results

### 3.1 Eruption sequence

True and false colour maps and ash BTDs were produced from the ABI data for the entire eruptive period (9 to 22 April 2021) using the instrument's high temporal resolution. An example from 10 April at 12:00 UTC is shown in Fig. 3. Animations showing the true and false colour images and the BTD maps for each eruptive event can be found at https://ora.ox.ac.uk/objects/ uuid:ca4e7a94-37c6-4d5f-b94a-1f287e661f8a (Taylor et al. 2023; webpage last accessed on 2 October 2023). These images have been carefully examined to get start and end times for each of the explosive events identified. The start and end times are reported in Table 4. The heights obtained with the BT method are reported in Table 5.

Using the ABI data, a minimum of 35 explosive events containing volcanic ash were identified. The number identified is limited by the temporal resolution of the instrument and cloud cover that might obscure the plume, and so could potentially be higher. For example, explosive events that last less than 10 minutes may not be identified in the 10-minute resolution data, or discrete eruptions or pulses that occur within the 1 or 10 minutes between images may be classified as the same eruption. Times where this is suspected are mentioned in Table 4. Additionally, lower level activity may not be visible within the ABI data. Figure 6a shows the number of explosive events which started on each day as determined with the ABI data. This is shown to be highest on the 10 April and decreases throughout the remainder of the eruptive sequence, with Fig. 6c showing that the repose time between explosive events increases over time. The duration of the eruptive events (Fig. 6b) is also shown to generally increase throughout the 9-22 April eruptive period as also noted by Joseph et al. (2022).

**Table 4.** Summary of plumes from La Soufrière seen with the ABI satellite instrument. The start and end times have been determined with the ABI images. These do not necessarily correspond to activity observed on the ground: the start time refers to when the plume is first visible in the ABI data, while the end time incorporates the time taken for the plume to rise and disperse away from the volcano. For a number of reasons discussed in section 2.2.2, the times presented here should be considered approximate. Due to the uncertainty with the end times, in some cases these are presented as a range. The eruption has been split into four phases based on the general character of the events. The phase numbers correspond to (1) initial explosive event; (2) sustained ash emission; (3) pulsatory phase; (4) waning phase.

| Phase | Event Number | Start Time (dd/mm/yyyy hh:mm UTC) | End Time (dd/mm/yyyy hh:mm UTC) | Notes |
|---|---|---|---|---|
| 1 | 1 | 09/04/2021 12:54 | 09/04/2021 17:14 – 17:44 | Small plume travelling W. Main optically thick plume travelling ENE, after 13:14 becomes less optically thick. Thick cloud cover after 15:34 makes end time difficult to determine |
| 2 | 2 | 09/04/2021 19:14 | 10/04/2021 04:44 – 4:54 | Large, sustained eruptive event with optically thick ash cloud. Shape of plume suggests possible strong pulses within this eruptive phase at around 19:24, 20:14, 21:14, 21:44, 22:24, 23:24 UTC on 9 April, and 03:34, 03:54 on 10 April. However, 10-minute temporal resolution does not allow these to be confirmed. |
| 3 | 3 | 10/04/2021 04:54 | 10/04/2021 06:04 – 06:34 | Beginning of shorter pulses. Optically thick ash with small gaps between |
| | 4 | 10/04/2021 06:54 | 10/04/2021 07:34 | Small emission to start followed by optically thick plume. |
| | 5 | 10/04/2021 07:44 | 10/04/2021 09:43 | 1-minute resolution data used here for first time in analysis (for the end time) |
| | 6 | 10/04/2021 09:44 | 10/04/2021 10:53 | |
| | 7 | 10/04/2021 10:54 | 10/04/2021 12:08 | |
| | 8 | 10/04/2021 12:09 | 10/04/2021 12:35 - 13:00 | |
| | 9 | 10/04/2021 13:01 | 10/04/2021 14:34 | |
| | 10 | 10/04/2021 14:35 | 10/04/2021 16:02 – 16:23 | Difficult to determine end time |
| | 11 | 10/04/2021 16:24 | 10/04/2021 18:47 | |
| | 12 | 10/04/2021 18:48 | 10/04/2021 19:56 – 21:25 | Difficult to determine end time |
| | 13 | 10/04/2021 21:26 | 10/04/2021 22:24 – 23:06 | |
| | 14 | 10/04/2021 23:07 | 10/04/2021 23:59 - 11/04/2021 00:37 | |
| | 15 | 11/04/2021 01:01 | 11/04/2021 01:41 – 02:49 | Difficult to determine end time |
| | 16 | 11/04/2021 02:50 | 11/04/2021 03:23 - 04:23 | |
| | 17 | 11/04/2021 05:05 | 11/04/2021 05:42 – 06:01 | |
| | 18 | 11/04/2021 08:02 | 11/04/2021 08:28 – 10:40 | Initially optically thick ash cloud becoming optically thin. Difficult to determine end time |

| Phase | Event Number | Start Time (dd/mm/yyyy hh:mm UTC) | End Time (dd/mm/yyyy hh:mm UTC) | Notes |
|---|---|---|---|---|
| | 19 | 11/04/2021 10:42 | 11/04/2021 12:00 – 12:08 | |
| | 20 | 11/04/2021 13:30 | 11/04/2021 14:52 – 18:09 | Initially optically thick ash cloud becoming optically thinner. Difficult to distinguish end time |
| | 21 | 11/04/2021 18:10 | 11/04/2021 19:08 – 20:09 | Initially optically thick ash cloud becoming optically thinner. Difficult to distinguish end time |
| | 22 | 11/04/2021 20:09 | 11/04/2021 20:50 – 23:44 | Initially optically thick ash cloud becoming optically thinner over time. Difficult to distinguish the start time from tail of previous event. |
| | 23 | 12/04/2021 00:09 | 12/04/2021 00:25 – 00:27 | Very faint plume to the E. |
| | 24 | 12/04/2021 00:46 | 12/04/2021 02:17 – 05:16 | Initially optically thick ash cloud becoming optically thinner over time. Hard to distinguish the end time |
| | 25 | 12/04/2021 08:06 | 12/04/2021 09:38 – 10:16 | Initially optically thick ash cloud becoming optically thinner over time. Difficult to distinguish the end time |
| | 26 | 12/04/2021 20:16 | 12/04/2021 20:36 – 20:55 | Very faint plume travelling to the E. Not confident this is volcanic |
| | 27 | 12/04/2021 21:13 | 12/04/2021 21:54 – 22:10 | Faint plume travelling to the E |
| 4 | 28 | 13/04/2021 10:36 | 13/04/2021 13:00 – 16:45 | Initially optically thick ash cloud becoming optically thin over time. Difficult to distinguish the end time |
| | 29 | 14/04/2021 00:53 | 14/04/21 02:32 | Small optically thin ash plume to the N. Faint emission makes it difficult to distinguish the start time. |
| | 30 | 14/04/2021 02:33 | 14/04/2021 05:38 – 07:19 | Difficult to determine end time |
| | 31 | 14/04/2021 15:41 | 14/04/2021 17:21 – 17:45 | Initially optically thick ash cloud becoming optically thinner. Becomes obscured by cloud |
| | 32 | 15/04/2021 08:58 | 15/04/2021 10:21 – 10:24 | Last use of 1-minute resolution data. Faint signal so hard to get the start time. Scene becomes cloudy. |
| | 33 | 16/04/2021 10:44 | 16/04/2021 13:34 – 17:04 | Optically thin plume visible amongst light cloud layers |
| | 34 | 18/04/2021 21:04 | 19/04/2021 00:04 – 00:44 | Optically thin plume. Difficult to determine start and end times |
| | 35 | 22/04/2021 15:34 | 22/04/2021 21:54 – 22:54 | Optically thin plume. Difficult to determine start and end times |

**Table 5.** Heights retrieved using Brightness Temperature (BT) method applied to ABI data.

| Event Number | Optically thick height solution (km) | | Plume centre height solution (km) | |
| --- | --- | --- | --- | --- |
| | Troposphere | Stratosphere | Troposphere | Stratosphere |
| 1 | $13.11^{+0.12}_{-0.12}$ | $22.77^{+0.23}_{-0.26}$ | $11.98^{+0.13}_{-0.16}$ | |
| 2 | $16.78^{+0.24}_{-0.20}$ | $17.45^{+0.11}_{-0.11}$ | $15.32^{+0.19}_{-0.24}$ | $19.20^{+0.15}_{-0.16}$ |
| 3 | $15.80^{+0.16}_{-0.18}$ | $18.89^{+0.14}_{-0.19}$ | $15.48^{+0.18}_{-0.20}$ | $19.20^{+0.16}_{-0.14}$ |
| 4 | $12.96^{+0.11}_{-0.11}$ | $21.91^{+0.19}_{-0.15}$ | $12.73^{+0.12}_{-0.12}$ | $22.28^{+0.28}_{-0.19}$ |
| 5 | $14.85^{+0.14}_{-0.15}$ | $20.01^{+0.14}_{-0.17}$ | $13.25^{+0.10}_{-0.10}$ | $21.36^{+0.16}_{-0.15}$ |
| 6 | $15.34^{+0.18}_{-0.16}$ | $19.16^{+0.19}_{-0.16}$ | $15.48^{+0.23}_{-0.18}$ | $19.03^{+0.16}_{-0.16}$ |
| 7 | $15.58^{+0.24}_{-0.19}$ | $18.95^{+0.16}_{-0.16}$ | $15.48^{+0.23}_{-0.16}$ | $19.02^{+0.16}_{-0.16}$ |
| 8 | $15.94^{+0.19}_{-0.22}$ | $18.71^{+0.15}_{-0.15}$ | $15.28^{+0.16}_{-0.14}$ | $19.22^{+0.13}_{-0.16}$ |
| 9 | $15.15^{+0.16}_{-0.15}$ | $19.30^{+0.09}_{-0.09}$ | $15.71^{+0.21}_{-0.23}$ | $18.93^{+0.15}_{-0.14}$ |
| 10 | $14.85^{+0.16}_{-0.15}$ | $19.38^{+0.10}_{-0.10}$ | $15.65^{+0.20}_{-0.21}$ | $18.90^{+0.12}_{-0.12}$ |
| 11 | $15.11^{+0.26}_{-0.21}$ | $19.19^{+0.15}_{-0.18}$ | $15.01^{+0.18}_{-0.16}$ | $19.24^{+0.12}_{-0.14}$ |
| 12 | $14.35^{+0.15}_{-0.12}$ | $19.91^{+0.23}_{-0.21}$ | $14.56^{+0.21}_{-0.15}$ | $19.61^{+0.21}_{-0.17}$ |
| 13 | $16.16^{+0.12}_{-0.13}$ | $18.44^{+0.17}_{-0.15}$ | $14.77^{+0.30}_{-0.24}$ | $19.67^{+0.26}_{-0.22}$ |
| 14 | $16.68^{+0.12}_{-0.11}$ | $17.70^{+0.12}_{-0.12}$ | $16.16^{+0.11}_{-0.12}$ | $18.35^{+0.16}_{-0.16}$ |
| 15 | $16.04^{+0.11}_{-0.12}$ | $18.51^{+0.18}_{-0.13}$ | $15.54^{+0.13}_{-0.13}$ | $19.42^{+0.22}_{-0.25}$ |
| 16 | $15.90^{+0.12}_{-0.11}$ | $18.46^{+0.13}_{-0.10}$ | $16.25^{+0.12}_{-0.12}$ | $18.15^{+0.11}_{-0.12}$ |
| 17 | $15.81^{+0.11}_{-0.10}$ | $17.94^{+0.12}_{-0.12}$ | $15.72^{+0.11}_{-0.10}$ | $18.05^{+0.12}_{-0.12}$ |
| 18 | $15.21^{+0.13}_{-0.13}$ | $19.22^{+0.13}_{-0.20}$ | $14.87^{+0.13}_{-0.13}$ | $19.56^{+0.12}_{-0.13}$ |
| 19 | $14.85^{+0.14}_{-0.13}$ | $19.49^{+0.15}_{-0.20}$ | $15.12^{+0.15}_{-0.15}$ | $19.09^{+0.22}_{-0.27}$ |
| 20 | $14.67^{+0.14}_{-0.14}$ | $19.46^{+0.10}_{-0.10}$ | $14.85^{+0.14}_{-0.13}$ | $19.48^{+0.11}_{-0.12}$ |
| 21 | $13.65^{+0.11}_{-0.11}$ | $21.54^{+0.26}_{-0.28}$ | $13.39^{+0.12}_{-0.12}$ | $21.99^{+0.29}_{-0.31}$ |
| 22 | $14.88^{+0.28}_{-0.23}$ | $19.56^{+0.31}_{-0.29}$ | $14.40^{+0.14}_{-0.13}$ | $20.13^{+0.22}_{-0.24}$ |
| 23 | $8.84^{+0.12}_{-0.12}$ | | | |
| 24 | $15.30^{+0.23}_{-0.43}$ | $19.86^{+0.19}_{-0.28}$ | $14.08^{+0.12}_{-0.12}$ | $20.66^{+0.13}_{-0.11}$ |
| 25 | $14.64^{+0.15}_{-0.13}$ | $19.61^{+0.06}_{-0.06}$ | $14.49^{+0.13}_{-0.12}$ | $19.68^{+0.06}_{-0.06}$ |
| 26 | $8.52^{+0.13}_{-0.13}$ | | $7.93^{+0.14}_{-0.16}$ | |
| 27 | $12.25^{+0.12}_{-0.12}$ | $24.27^{+0.30}_{-0.26}$ | $10.44^{+0.14}_{-0.15}$ | |
| 28 | $14.42^{+0.22}_{-0.19}$ | $19.58^{+0.11}_{-0.08}$ | $14.09^{+0.16}_{-0.13}$ | $19.79^{+0.11}_{-0.11}$ |
| 29 | $5.93^{+0.18}_{-0.18}$ | | $2.40^{+0.18}_{-0.17}$ | |
| 30 | $13.84^{+0.11}_{-0.12}$ | $20.75^{+0.22}_{-0.21}$ | $13.53^{+0.12}_{-0.12}$ | $21.32^{+0.21}_{-0.21}$ |
| 31 | $11.02^{+0.14}_{-0.14}$ | | $13.51^{+0.10}_{-0.10}$ | $21.59^{+0.53}_{-0.92}$ |
| 32 | $7.80^{+0.14}_{-0.14}$ | | $6.81^{+0.19}_{-0.19}$ | |
| 33 | $5.32^{+0.23}_{-0.20}$ | | $3.71^{+0.19}_{-0.19}$ | |
| 34 | $5.81^{+0.19}_{-0.17}$ | | $3.66^{+0.17}_{-0.15}$ | |
| 35 | $9.11^{+0.13}_{-0.13}$ | | $3.20^{+0.21}_{-0.26}$ | |

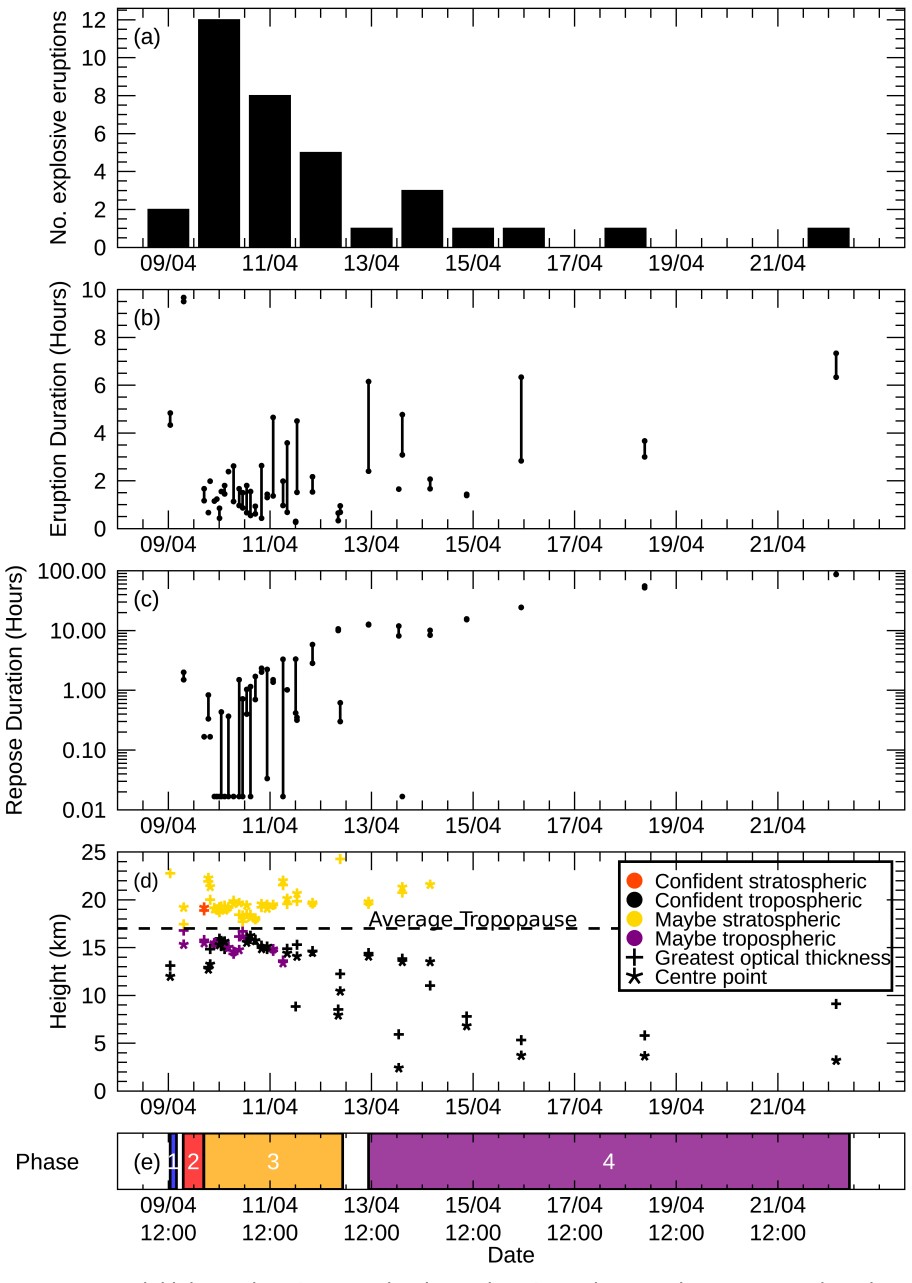

**Figure 6.** Summary of ABI results. (a) Number of explosive eruptions on each day. (b) Duration of each explosive event. (c) Repose duration between events. (d) Heights retrieved with optically thick and centre BT methods. (e) Using the ABI data the 2021 eruption has been divided into the four phases with the timing shown here.

The eruptive events identified from the ABI data have been split into four phases each with distinct characteristics. The first emission (and first phase of the eruption), is likely to correspond to arrival of gas-rich magma at the surface, and loss of the lava-dome cap. This is seen with ABI at around 12:54 UTC (08:54 LT) on 9 April, and corresponds well with the 08:41 LT (12:41 UTC) eruption start time stated in Joseph et al. (2022). Using the ABI data, this emission appears to be relatively short lived, lasting around 4.3 to 4.8 hours, finishing at around 17:14-17:44 UTC (although significant cloud cover makes the end time difficult to distinguish). During this time two plumes are evident: a low altitude plume can be seen travelling to the west from the volcano, while the main plume travels to the east-north-east, with a height of around $13.11^{+0.12}_{-0.12}$ km based on the optically thick BT method. This is greater than the 8 km reported by the Belmont Observatory (Global Volcanism Program, 2021b). However, differences between ground based and satellite measurements can arise for a number of reasons including the different viewing angles and assumptions, poor visibility on the ground, the plume rising too high for the height to be accurately obtained from the ground and different definitions of plume height (e.g. Tupper et al., 2004; Tupper and Kinoshita, 2003; Tupper and Wunderman, 2009).

Following a short pause of around 1.5 to 2 hours, the second phase of the eruption started at around 19:14 UTC on 9 April. This is characterised by a sustained ash emission event with a duration of around 9.5 to 9.67 hours. Height estimates based on the BT method are between 15.32 and 19.20 km. In this case, the plume direction and wind profile do not help determine whether the tropospheric or stratospheric solution is more likely. The structures of the plume suggest that there may have been some strong pulses within this sustained phase (see Table 4) but these cannot be easily resolved from the 10-minute ABI images. The observations using the ABI instrument are generally consistent with those made by Joseph et al. (2022) who mention a "sustained" and "pulsing" explosive phase starting at 16:00 UTC on the 9 April and finishing at 6:00 UTC on 10 April.

The explosive events identified with the ABI instrument suggest that the eruption entered a third phase at 4:54 UTC on 10 April. This phase appears more pulsatory with a further 25 discrete ash emission events identified within a 65.3 hour period. The use of the 1-minute ABI data during this phase greatly helped to identify the event timings: highlighting the considerable advantage of this instrument. Joseph et al. (2022) characterise this phase as one of "ash venting". The length of these explosive events identified from the ABI data varied from around 16 minutes to up to 4.65 hours. This style of repetitive ash bursts is similar to the "Vulcanian" explosions documented at multiple andesite volcanoes, which is considered to reflect the rapid sealing, pressurisation and failure of magma within the conduit (e.g. Druitt et al., 2002; Watt et al., 2007; Joseph et al., 2022). The first pulse is shown to most likely be in the stratosphere with height solutions of $18.89^{+0.14}_{-0.19}$ and $19.20^{+0.16}_{-0.14}$ km for the optically thick and centre BT methods respectively. The majority of the remaining pulses in this phase have been labelled as most likely tropospheric or uncertain. The height results for each event are shown in Table 5.

The fourth and final phase of the eruption, as defined with the ABI data, began on 13 April at 10:36 UTC and continued to the 22 April. During this phase there were a further eight distinct events each lasting a few hours, with generally longer repose times than the previous phases, and with each event emitting ash into the troposphere (heights between $2.4^{+0.18}_{-0.17}$ and $14.42^{+0.22}_{-0.19}$ km). Fig. 6d shows that during this phase, the heights for each eruptive event fall suggesting a decrease in eruptive power. Compared to the earlier phases, the RGB images suggest that the plumes in this waning stage are more optically thin

and it is possible, especially for events 29 and 33-35, that the heights obtained are underestimates due to upwelling radiation from beneath the plume. A number of the explosive events in this phase appear to have an explosive eruption which produces a small optically thick cloud, followed by a longer emission of a less optically thick ash cloud, while the last three events consist of small plumes with ash that is optically thin. The final eruptive event emitted ash into the lower troposphere for around 6.3-7.3 hours.

In contrast to this study, Horváth et al. (2022) count 49 explosive events between 9 and 22 April using the ABI data. The higher number reported by Horváth et al. (2022) is partly because they have divided event 2 (as termed in this paper) into multiple explosive events. Sparks et al. (2024) also document the sequence of eruptive events occurring during the April 2021 eruption. Their analysis is primarily based on seismic data, but they also make note of the first observation times with the ABI instrument. As expected, there are differences between the start times identified with the seismic and ABI instruments, as there is some time before the plume becomes visible with the satellite instrument. On the whole, their ABI observation times (only start time reported) agree well with the ones presented here, however there are a few differences. Firstly, there are differences in the timings of the events: they used the 10-minute full disc rather than the meso-scale data, and other differences may arise from the subjective nature of this exercise. Secondly, this study identifies some events which are not included in Sparks et al. (2024) timeseries, while Sparks et al. (2024), using the seismic data, is able to split event 2 (as referred to here) into multiple events. Finally, Sparks et al. (2024) also identify four phases to the eruption, and while there is some overlap with the phases outlined here, there is disagreement with the timings: reflecting the different datasets and metrics used.

## 3.2 SO$_2$ dispersion

The linear and iterative SO$_2$ retrievals have been applied to IASI spectra from 9 to 30 April 2021. The iterative retrieval column amounts for 9 to 26 April are displayed in Fig. 7. Figure 8 shows examples of the column amount and height outputs from the iterative retrieval for 10 and 11 April. Animations of the iterative retrieval column amount and height outputs for the full time period explored can be found in the supplementary material along with animations showing the error outputs. The mapped results have been obtained by gridding the outputs for all three IASI instruments to a regular 0.125 by 0.125 degree grid.

Figure 7 shows the SO$_2$ column amounts from the iterative retrieval. In these maps it is possible to see the evolution of the La Soufrière SO$_2$ plumes between 9 and 26 April. The plume is first apparent as a faintly elevated emission to the east of the volcano in the ascending overpass on 9 April 2021 (incorporates ascending orbits from 12:00 9 April to 12:00 10 April). A stronger signal is then visible in the descending orbits on 10 April fanning out to the east of the volcano across the North Atlantic. The general east and south-eastward transport of the plume between 9 and 11 April implies that the bulk of SO$_2$ has been emitted into the troposphere, with Fig. 4a indicating wind directions between $\sim$90-140° dominating in the troposphere between 8 and 17 km. By the ascending orbits on 11 April (incorporates ascending orbits from 12:00 11 April to 12:00 12 April), the plume has travelled across the Atlantic and has almost reached the west coast of Africa. The higher values for the column amounts leading back to the source reflect the frequent eruptive events that occurred over this time period. The general eastward transport of the plume is consistent with observations by Babu et al. (2022) who tracked the plume with the Ozone Monitoring Instrument. On the 12 April, while the bulk of the plume is still advancing towards the east (Fig. 7), a fraction of

the plume travels to the south and west of the volcano. The wind directions shown in Fig 4a, imply that for south or westward transport, either some $SO_2$ has been emitted into the lower parts of the troposphere (less than 5 km) or more likely that some $SO_2$ has been emitted, or has been lofted, into the stratosphere. Over the next few days the plume continues to be transported both to the east and west of the volcano and by the 21 April the plume has been transported around the circumference of the globe. Note that there is an emission of $SO_2$ from an eruption at Sangay in Ecuador from 12 April (Global Volcanism Program, 2021d). This combines with the plume from La Soufrière and the two cannot be easily distinguished from each other from 13 April.

Figure 9b shows the distribution of the $SO_2$ mass with latitude. The largest concentration of $SO_2$ (up to 17 April) is between 0 and 20° N but parts of the plume have reached 45° north and south. This highlights the ability of tropical eruptions to transport $SO_2$ across both hemispheres, rather than being confined to a narrower latitude band like plumes from eruptions at high latitudes (Schmidt and Black, 2022). Also highlighted in Fig. 9, is a line of elevated $SO_2$ around 15° S. This is the latitude of Sabancaya volcano (71.857°W, 15.787° S, summit elevation of 5.96 km) in Peru which was erupting throughout the studied time period. Plumes from Sabancaya can be distinguished in the iterative retrieval maps in the supplementary material.

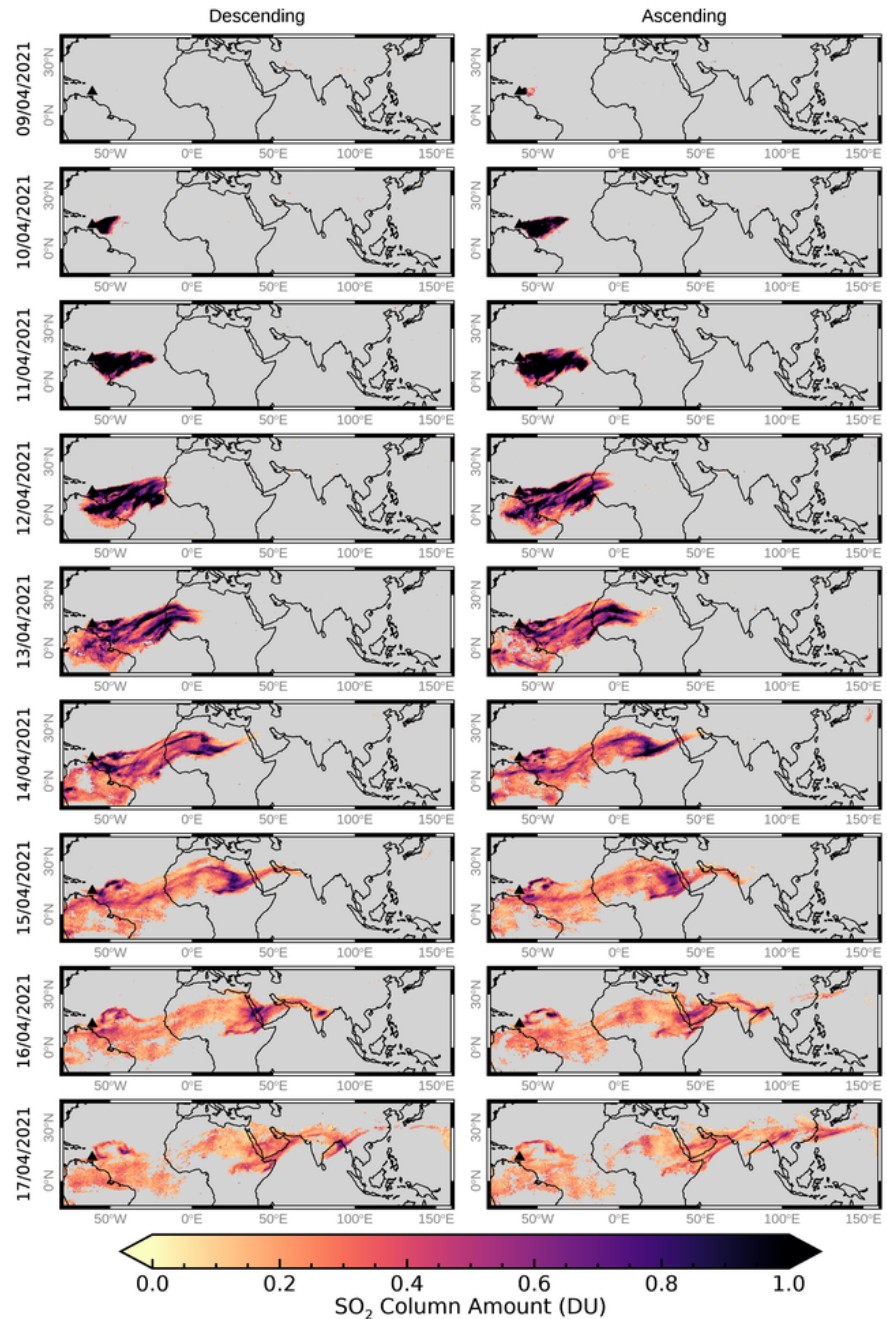

**Figure 7.** Maps showing the gridded IASI iterative SO$_2$ retrieval column amount output for 9 to 26 April 2021. Maps on the left show a composite of the descending orbits (09:30 LT) from 00:00 to 23:59 UTC on each day, and the maps of the right show a composite of the ascending orbits (21:30 LT) from 12:00 on the day stated to 12:00 of the following day. Note that from 18 April, the geographic region displayed has been expanded to show the full extent of the plumes on these days. The location of the volcano is marked by a black triangle.

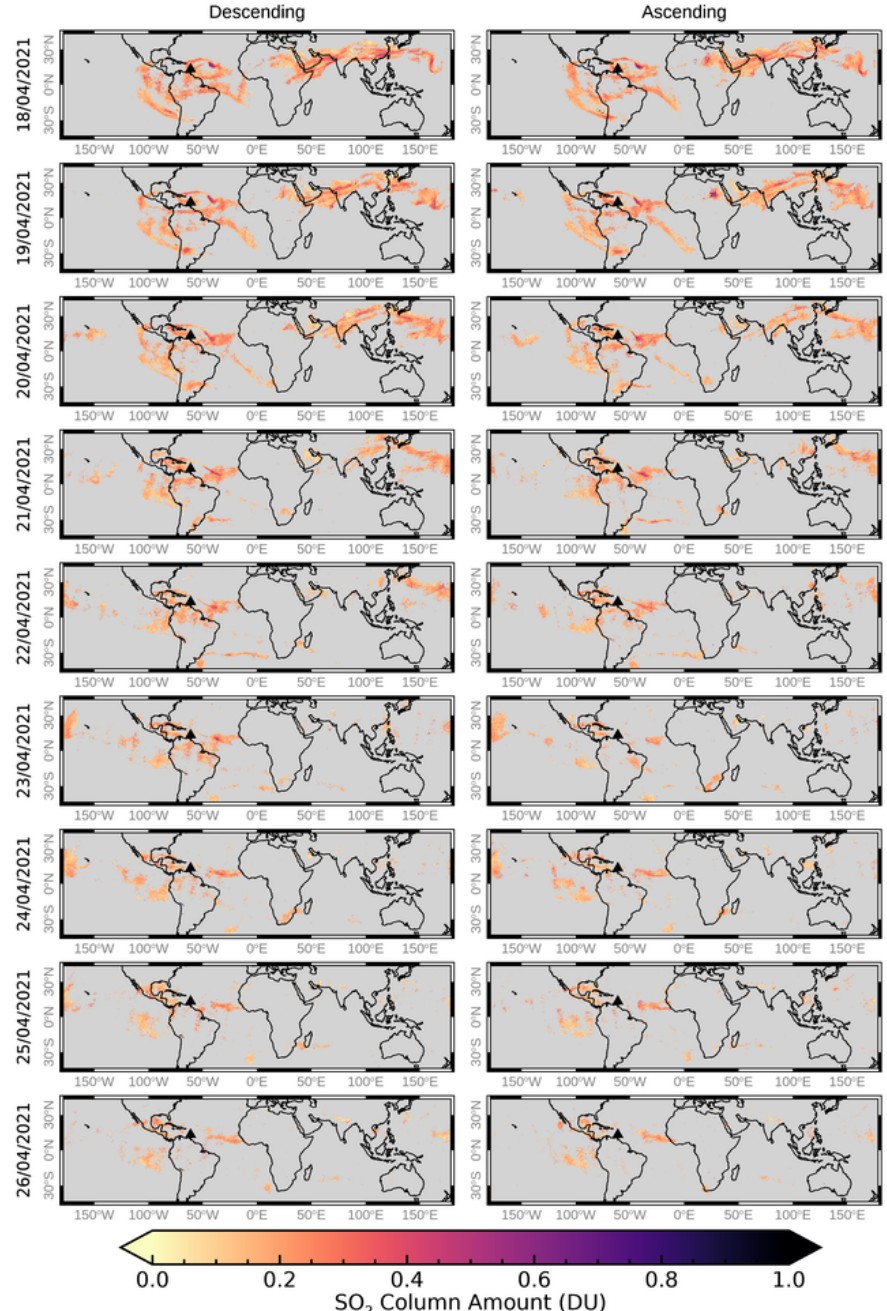

**Figure 7.** *Continued.*

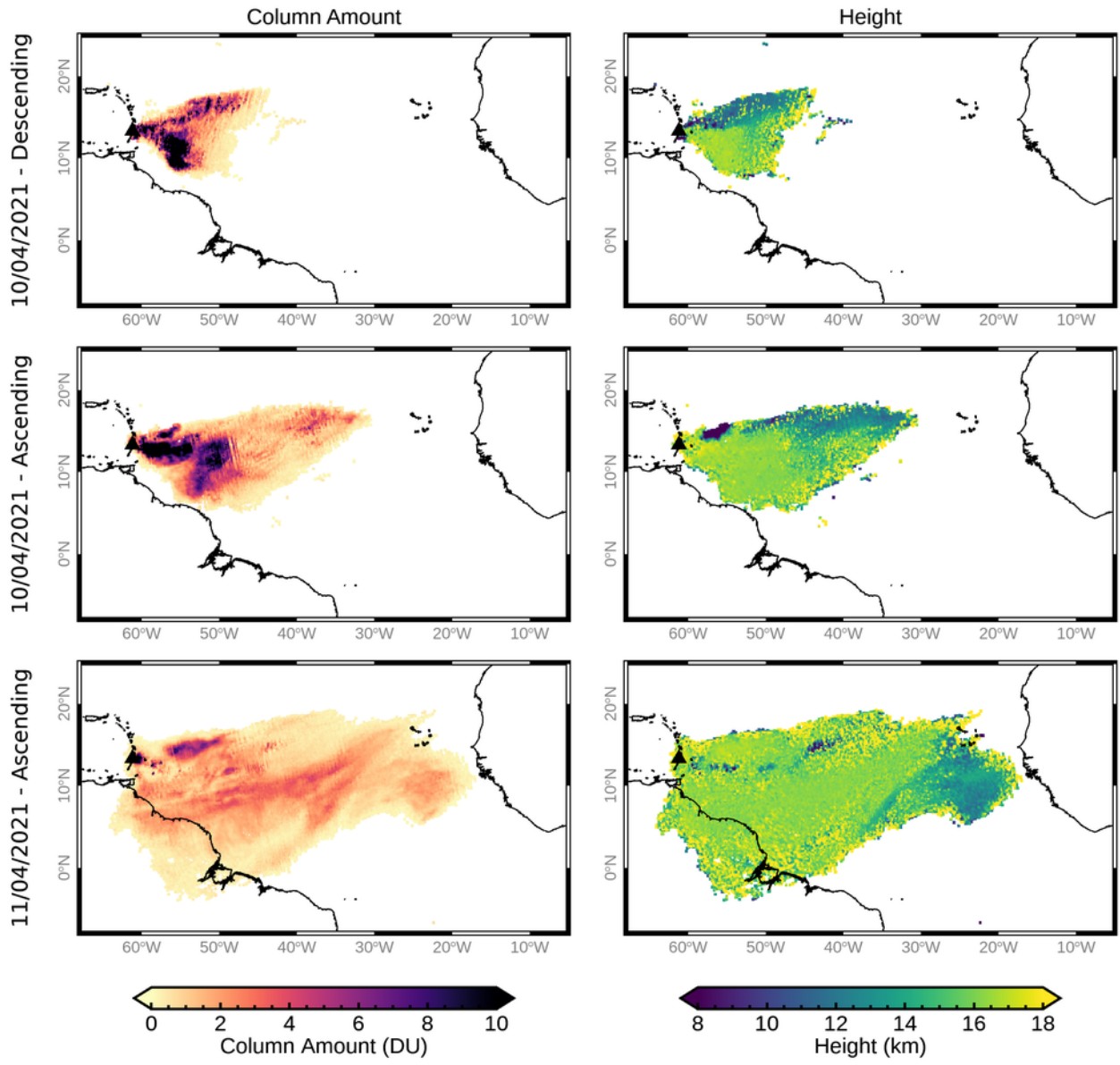

**Figure 8.** Examples of the IASI SO$_2$ iterative retrieval gridded column amounts and heights for scenes from the 10 and 11 April 2021. The date and orbit direction are indicated on the left-hand side of the figure. Maps of the results for the entire time period studied can be found in the supplementary information.

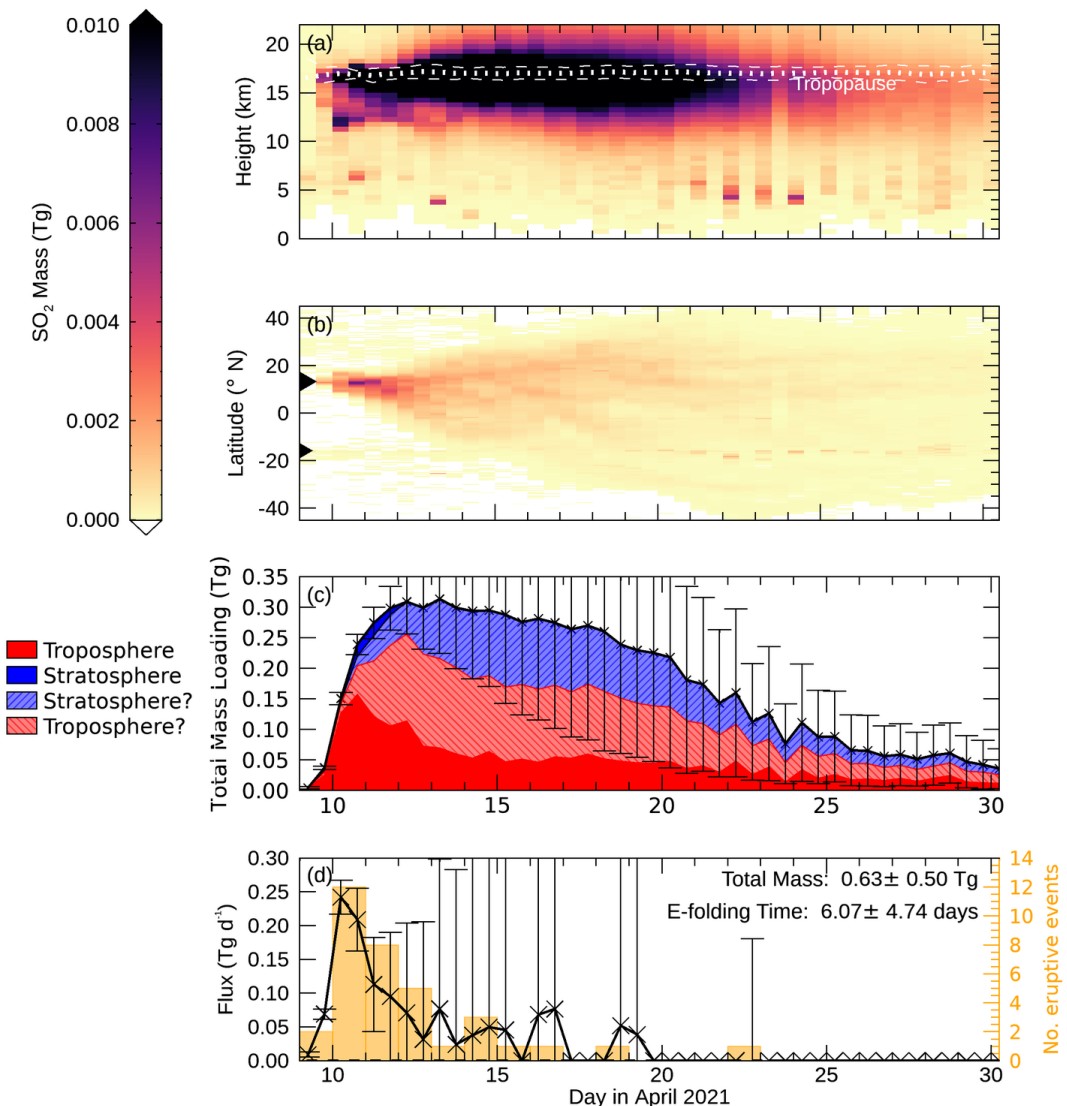

**Figure 9.** Summary of the IASI $SO_2$ results generated from iterative retrieval outputs for 9 to 30 April 2021. Each of these has been computed for the -45 to 45° N and -180 and 180° E region. (a) Vertical distribution of the $SO_2$ mass across the plume with time. The dashed lines show the mean height of the tropopause and one standard deviation from this; (b) Distribution of the $SO_2$ mass with latitude. The black triangle at 13.33° N indicates the latitude of La Soufrière. The smaller triangle at 15.79° S shows the latitude of Sabancaya in Peru which also erupted during the studied time period; (c) Timeseries of the total mass of $SO_2$ with errors. Using the retrieved $SO_2$ heights and the tropopause heights this has been divided into tropospheric and stratospheric $SO_2$. The height errors have also been used to create two more categories: heights which are stratospheric but given the height error could be tropospheric, and heights which are tropospheric but could be stratospheric. (d) The flux of $SO_2$ with time and the number of eruptive events identified from the ABI data on each day.

### 3.3  SO$_2$ mass and flux

A total mass timeseries, derived from the IASI iterative retrieval output for the -45 to 45° N and -180 and 180° E region, is shown in Fig. 9c. Note that other volcanoes (e.g. Sabancaya in Peru; Sangay in Ecuador) were erupting at this time and any SO$_2$ from these eruptions entering the region will affect the total mass, e-folding and flux estimates. However, given these are smaller emissions than La Soufrière, their impact is negligible and within the reported errors. The IASI derived SO$_2$ mass loading peaks at $0.31 \pm 0.09$ Tg in the descending orbits on 13 April. This is fairly small compared to other eruptions studied by IASI including Nabro (1.6 Tg), Kasatochi (0.9 Tg), Grímsvötn (0.75 Tg), Copahue (0.72 Tg) and Sarychev (0.6 Tg) in 2011, 2008, 2012 and 2009 respectively (Carboni et al., 2016). It is more similar to the maximum mass loading recorded from Alu Dalafila (0.2 Tg) in November 2008 (Carboni et al., 2016).

The fact that the total SO$_2$ mass loading for La Soufrière peaks a few days after the 2021 eruption began could be linked to the multiple explosive events which occurred between 9 and 13 April leading to an increase in the total atmospheric SO$_2$ loading: SO$_2$ emitted at a rate greater than the existing SO$_2$ declines. The Global Volcanism Program (2021c) reported an early satellite estimate of the SO$_2$ mass of 0.4 Tg on 10 April: larger than the estimate from IASI on the same day $0.24 \pm 0.02$ Tg, ascending result). Differences between instruments and retrievals are expected due to the varying sensitivities to SO$_2$, different height assumptions, the differing effects of volcanic ash and aerosols, different spatial resolutions and observation times. Carn et al. (2016) for example notes that there are often differences between the results from UV and infrared sensors. As mentioned in section 2, the IASI iterative SO$_2$ retrieval column amounts, and subsequently the total masses, are affected by volcanic ash and instrument sensitivity and so the column amounts and total mass values presented here should be considered minimum estimates.

From 13 April the total mass of SO$_2$ is shown to fall: as the SO$_2$ is removed from the atmosphere through deposition, by conversion to sulfate aerosol, or dilution below the detection limit of the instrument. The e-folding time used here (see section 2.1.3) describes this loss process. This varies with a number of factors including the latitude of the volcano, the injection height of the plume, meteorological conditions, cloud cover, water vapour, season and the presence of ash (Carn et al., 2016; Zhu et al., 2020; Schmidt and Black, 2022; Zhu et al., 2022). Typically, the e-folding time varies from hours to days in the lower troposphere to weeks in the stratosphere. A first estimate of 5.47 days for the SO$_2$ e-folding time was estimated by fitting Eq. 1 to the IASI total SO$_2$ masses between 23 to 30 April 2021. Following the application of the Carboni et al. (2019) method to compute the flux and average e-folding time, the average e-folding time was adjusted to $6.07 \pm 4.74$ days. This is in line with other eruptions including Jebel at Tair (2007) and Merapi (2010), both volcanoes in the tropics and which emitted plumes between 15 and 18 km, and had e-folding times of between 2 and 4 days (Carn et al., 2016). Given that there was ash emission during the La Soufrière eruption, it is possible that the e-folding time could have been reduced as a result of accelerated oxidation of SO$_2$ due to reactions on the ash surface, as was seen for the Kelut eruption in a study by Zhu et al. (2020). Some of the SO$_2$ emitted during the La Soufrière eruption was converted to sulfate aerosol, as is shown in Babu et al. (2022) and Bruckert et al. (2023).

**Table 6.** Number of eruptive events identified with ABI and IASI SO$_2$ fluxes for 9 to 22 April 2021. Note that these estimates may be affected by other volcanoes which erupted during this period.

| Date (dd/mm/yyyy) | Number of Events | IASI SO$_2$ flux (Tg d$^{-1}$) | |
| --- | --- | --- | --- |
| | | Descending (00:00 to 23:59 UTC) | Ascending (12:00 to 12:00 day+1 UTC) |
| 09/04/2021 | 2 | 0.01 ± 0.00 | 0.07 ± 0.01 |
| 10/04/2021 | 12 | 0.24 ± 0.02 | 0.21 ± 0.05 |
| 11/04/2021 | 8 | 0.11 ± 0.07 | 0.10 ± 0.09 |
| 12/04/2021 | 5 | 0.07 ± 0.13 | 0.03 ± 0.17 |
| 13/04/2021 | 1 | 0.08 ± 0.22 | 0.02 ± 0.26 |
| 14/04/2021 | 3 | 0.04 ± 0.28 | 0.05 ± 0.31 |
| 15/04/2021 | 1 | 0.05 ± 0.31 | 0.00 ± 0.00 |
| 16/04/2021 | 1 | 0.07 ± 0.36 | 0.08 ± 0.35 |
| 17/04/2021 | 0 | 0.00 ± 0.00 | 0.00 ± 0.00 |
| 18/04/2021 | 1 | 0.00 ± 0.00 | 0.05 ± 0.37 |
| 19/04/2021 | 0 | 0.04 ± 0.35 | 0.00 ± 0.00 |
| 20/04/2021 | 0 | 0.00 ± 0.00 | 0.00 ± 0.00 |
| 21/04/2021 | 0 | 0.00 ± 0.00 | 0.00 ± 0.00 |
| 22/04/2021 | 1 | 0.00 ± 0.00 | -0.04 ± 0.22 |

The SO$_2$ masses from IASI were converted to flux estimates following the method described in section 2.1.3 (note that underestimates in the total masses may affect these results). The daily SO$_2$ fluxes computed from the IASI results are shown in Fig. 9d along with the number of explosive events which started on each day as determined with ABI data. Both are also reported in Table 6. There is generally a good correspondence between the number of eruptive events and the SO$_2$ flux. The greatest flux ($0.24 \pm 0.02$ Tg d$^{-1}$) from the IASI data occurs on the 10 April in the descending overpasses which incorporates the second eruptive phase (sustained emission), and the beginning of the pulsatory phase. Following this the SO$_2$ flux derived from IASI data is shown to decrease over the third eruptive phase which ends on the 12 April. The flux remains low for the final phase of the eruption and no emission is visible following the eruption on 22 April. These fluxes agree reasonably well with the flux range reported in Joseph et al. (2022) which varied between 2.76 x 10$^5$ t d$^{-1}$ (0.276 Tg d$^{-1}$) on 10 April to 331 t d$^{-1}$ (0.0003 Tg d$^{-1}$) on 22 April.

By summing the retrieved IASI SO$_2$ fluxes and multiplying by the time step between the images (assumed to be 12 hours) it is estimated that in total this eruption emitted $0.63 \pm 0.5$ Tg of SO$_2$. The error is likely to be overestimated due to the way in which the total mass errors are computed (see section 2.1.2). An alternative way of computing the minimum and maximum total emitted SO$_2$ is to add/subtract the flux and error for each time step, and multiple by the time step, rather than sum the errors in quadrature. This gives a minimum and maximum estimates of the total SO$_2$ mass are 0.24 Tg and 2.43 Tg respectively.

## 3.4 SO$_2$ plume heights

Examples of the iterative retrieval height results are shown in Fig. 8. The heights in the descending orbits on 10 April are variable. The heights in the northern edge of the plume (mostly 10 to 13 km, some less than 10 km) are lower than the rest of the plume. The southern part of the SO$_2$ plume is higher: primarily between 13 and 18 km, and in some cases exceeding this.

Similar structures can be seen in the following few days. There is no obvious gradient to the heights in the upper part of the plume as can be seen in IASI SO$_2$ height measurements shown in the supplementary material of Koukouli et al. (2022) for 10 and 11 April (using the Clarisse et al., 2014 method). Koukouli et al. (2022) show heights increasing from south to north of the plume: which matches well with the wind directions shown in Fig. 4a. The IASI retrieval used here relies on temperature and water variations in the atmosphere, which do not vary as significantly around the tropopause, which may affect the results.

Multiple retrieval setups were explored (including varying the retrieval first guess height and varying the plume thickness) but the results were similar in each case. Nevertheless, there is a broad agreement with Koukouli et al. (2022) which reports average heights of $15.7 \pm 1.16$ km for IASI using Clarisse et al. (2014) retrieval and $14.94 \pm 3.87$ km from TROPOMI (Hedelt et al., 2019 method) (note that these averages are based on a subset of the plume). Additionally, there is agreement with the 13 to 15 km injection heights obtained by Esse et al. (2024) using backwards trajectory modelling. The IASI SO$_2$ heights are

also generally consistent with ash height estimates reported in Global Volcanism Program (2021b) bulletin report; although it should be noted that SO$_2$ and ash are not always colocated spatially, temporally or in height (e.g. Holasek et al., 1996b; Thomas and Prata, 2011; Moxnes et al., 2014; Prata et al., 2017). The IASI results presented here are similar to observations made by the Multi-angle Imaging Spectroradiometer (MISR) on-board the Terra satellite (Yue et al., 2022). Results from this instrument on 10 April showed that there are multiple layers of ash, with two primary layers at 12 and 18 km, and some parts of the plume

reaching 20 km (Yue et al., 2022). Horváth et al. (2022) analysed 30 of the eruptive events using the ABI sideview and found that most of the plumes rose to around the height of the tropopause (16 to 17 km) or entered the lower stratosphere (18 to 20 km). They also reported the heights of the overshooting tops which went up to 23 km for the largest explosive events. Finally, there is agreement the IASI SO$_2$ heights here and those from the Cloud-Aerosol Lidar with Orthogonal Polarization (CALIOP) and ICON-ART model which show a volcanic plume layer between 15 and 20 km on 13 April (Bruckert et al., 2023).

The IASI results show that the structures within the plume evolve as the plume stretches across the Atlantic with the higher values (15 to 17 km) making up the majority of the image. Note that over time there appears to be more speckle in the height results (see animation in the supplementary files) which reflects increased uncertainty. The variations in height across the entire SO$_2$ plume are likely to be linked to the multiple injection heights and pulses which occurred during the earlier eruptive events, as discussed in section 3.1. The lower heights obtained from the ABI data and mentioned in the Global Volcanism Program

(2021b) bulletin report, for the later stages of the eruption (e.g. 14-22 April), are not reflected in the IASI results. This is potentially because the retrieval has been setup in a way that is optimised for higher plumes, the fact that these eruptive events emitted less SO$_2$ (below the detection limit of the retrieval), or due to multiple layers of SO$_2$.

The evolution of the vertical distribution of the SO$_2$ mass across the plume is shown in Fig. 9a. This is computed from the gridded column amounts (converted to mass values) and the gridded heights for each day (descending and ascending). On 10

April, $SO_2$ is shown to be emitted at multiple heights: a smaller emission between 11 and 13 km and a larger quantity between 15 and 18 km. The lower layer largely disappears within three days. Between 12 and 18 April the $SO_2$ mass is shown to be spread over 13 to 19 km. Lines showing the average tropopause height and one standard deviation from this are displayed in Fig. 9a. This was estimated based on the World Meteorological Organisation's (WMO) definition of the tropopause: that the tropopause occurs at the lowest level where the temperature lapse rate falls beneath 2 K km$^{-1}$ for at least 2 km. Note that this is computed from the tropopause heights for the pixels containing $SO_2$ and where the iterative retrieval output passes the quality control outlined in section 2.1.2: this means that the number of pixels used in this calculation varies over time. Much of the $SO_2$ appears to be in the upper troposphere and around the height of the tropopause which supports the observation made by Carboni et al. (2016) that $SO_2$ from explosive eruptions often ends up at the tropopause. This observation is consistent with the direction of travel of the plume (to the east) and the wind direction at these heights (which has an easterly direction in the upper troposphere and around the tropopause, Fig. 4a). A fairly persistent lower layer of $SO_2$ can be observed between 3 and 7 km. Some of this can be attributed to emissions at Sabancaya volcano in Peru. From 19 April a layer can also be observed at 14 to 14.5 km. This is around the height of the *a priori* and so is most likely due to a loss of height information in the IASI spectra as the plume thins.

Figure 9c shows the fraction of $SO_2$ which has been categorised as either tropospheric or stratospheric. Note that this has been calculated on a pixel by pixel basis using the height of the tropopause. Also shown in this plot, is the degree of uncertainty with regard to the height of $SO_2$ relative to the tropopause. Figure 9c highlights that a fraction of the $SO_2$ mass which while classified as stratospheric could in fact be tropospheric when the height errors are considered. Similarly it shows that the fraction of the mass classified as tropospheric which could be stratospheric. In general it appears that there is a larger amount of $SO_2$ in the troposphere. For example, in the descending orbits on 13 April, 68.88 % and 31.12 % of the $SO_2$ was shown to be in the troposphere and stratosphere respectively. However, based on the error bars, 67.78 % of the tropospheric result could also be classed as stratospheric given the overlap of the error bars with the tropopause height, and 98.65 % of the $SO_2$ classed as stratospheric could be tropospheric.

## 4   Comparison with the 1979 eruption

There are a number of similarities between the 1979 and 2021 eruptions at La Soufrière. Shepherd et al. (1979) give a detailed account of the eruptive sequence for the 1979 eruption and report a series of explosive eruptions between 13 and 26 April 1979, with the first explosions occurring from within a water-filled summit crater lake. In the first few hours after the eruption began on the 13 April 1979 (9:30 to 15:00 UTC), continuous emissions of steam were punctuated with explosive eruptions producing ash clouds that reached greater than 8 km (Shepherd et al., 1979). This activity was followed by a series of six explosive events with varying durations between 20:05 on 13 April and 15:50 on 14 April. Estimates from satellite data on 13 and 14 April give ash heights of 17 to 18 km for two of these plumes (Shepherd et al., 1979). Following this, between 14 and 17 April, the volcano entered a new phase with only occasional ash emissions rising 1 to 2 km above the vent. Another explosive event occurred on the 17 April with ash emissions reaching 18.7 km. Following this, the intensity of activity generally declined, although two

more explosive events occurred on the 22 and 26 April (Shepherd et al., 1979). By the end of the 1979 explosive sequence, the crater had been infilled with pyroclastic debris, and the summit lake had disappeared. The 2021 explosive eruption sequence shares a number of similarities with the 1979 eruption, even though the early stages of the 2021 eruption were rather different: with 3 months of dome effusion into a "dry" summit crater. Both the 1979 and 2021 eruptions consisted of a series of explosive events and both had a pulsatory phase in the first few days after the start of the eruption: although notably a larger number of discrete events occurred during the 2021 eruptive period. The later eruptive events in both the 1979 and 2021 eruptions also had a longer repose time.

Another similarity between the 1979 and 2021 eruptive periods are the plume heights. Measurements of the 1979 heights were far sparser: with estimates coming from relatively infrequent satellite data, ground/ship based observations and aircraft observations. As mentioned above, Shepherd et al. (1979) gives height estimates for individual eruptive events ranging between 8 and 18.7 km, similar to the heights reported here for the April 2021 eruption. This is similar to this study which reports a fairly wide range of heights for the 2021 eruption. Krueger (1982) used SMS-1 satellite data to estimate that part of the plume exceeded 18 km: placing it in the stratosphere. Another study using an airborne LiDAR by Fuller et al. (1982) showed layers of ash between 16 and 19.5 km. These heights for ash are in a similar range to those for $SO_2$ reported in this study, although again the difficulty of using ash as a proxy for $SO_2$ is noted. While both Krueger (1982) and Fuller et al. (1982) indicate that parts of the plume reached the stratosphere it is possible that there was significant variability across the full extent of the plume which was unmeasured, and so it is difficult to quantify how much of the plume was tropospheric and stratospheric for the 1979 eruption.

A significant difference between the two eruptions is the quantity of $SO_2$ observed by satellite instruments. The 1979 eruption of La Soufrière is one of the first observations of volcanic $SO_2$ made by the Total Ozone Monitoring Instrument (TOMS) but elevated emissions were only seen on 2 days (Carn et al., 2003). In total only ∼3 kt of $SO_2$ was measured with TOMS (Carn et al., 2016): substantially lower than the estimates reported here from IASI for the 2021 eruption. Carn et al. (2016) indicates that the TOMS $SO_2$ estimate is only a fraction of the total emitted with Scaillet et al. (2003) obtaining a total sulfur mass of 0.46 Tg based on petrological methods (∼ 1 Tg $SO_2$; Carn et al. 2016). Carn et al. (2016) suggests that the majority of sulfur emission may have been in the form of $H_2S$, with slow oxidation preventing $SO_2$ detection with TOMS. It is also possible that part of the low $SO_2$ mass estimate from TOMS could be attributed to the lower resolution of the instrument compared to the satellite instruments observing the 2021 eruption.

# 5  Conclusions

This study has used satellite data to gain insights into the chronology of events, and the various emissions of ash and $SO_2$ during the April 2021 eruption of La Soufrière. Using data from ABI it is possible to identify the emission of ash, and in this study the instrument's high temporal resolution (up to 1 minute) was used to identify and characterise different phases of the April 2021 La Soufrière eruption. Careful examination of true and false colour images, and BTD maps, allowed the identification of the start and end times of 35 explosive events that took place during the eruption: although this should be

considered a minimum estimate. Based on these observations four eruptive phases have been identified: (1) an initial explosive event producing a ash cloud; (2) a strong and sustained ash emission lasting several hours and producing a large optically thick ash cloud; (3) a pulsatory phase with 25 discrete explosive events in a 65.3 hour period, varying in duration between 16 minutes and 4.65 hours, each producing optically thick ash clouds; (4) a waning phase with 8 eruptive events with lower plume heights, optically thinner ash clouds and greater durations between explosive events.

Measurements with retrievals developed for the IASI satellite instrument were able to track the plumes of $SO_2$ from the 2021 La Soufrière eruption for several weeks after the first eruption occurred. A peak value of $0.31 \pm 0.09$ Tg was obtained for the total $SO_2$ mass burden on 13 April. However, it is likely that this is an underestimate due to the effects of volcanic ash on the results and the presence of $SO_2$ below the detection limit of the instrument. Using the daily IASI $SO_2$ fluxes it was possible to estimate the total emission of $SO_2$ to the atmosphere which was $0.63 \pm 0.5$ Tg with an upper and lower limit of 2.43 and 0.24 Tg respectively (considering the full extent of the uncertainties on each day). Again, due to the potential underestimation of the mass, these should be considered minimum estimates. The retrieved heights show interesting structures in the plume that reflect the multiple eruptive pulses: consistent with observations on the ground and made by other satellite instruments. The height results suggest that the majority of $SO_2$ was emitted into the upper troposphere and lower stratosphere, with most of the gas being concentrated between 13 and 19 km.

The 2021 eruption showed some remarkable similarities with the 1979 eruption both in terms of the height and the eruptive sequence. This emphasises the importance of studying these eruptions, utilising as many data sources as possible, in order to better understand the volcano and the character of its eruptions so as to be better prepared for future eruptive events (e.g. Joseph et al., 2022; Barclay et al., 2022).

*Data availability.* GOES-16 satellite data are available through NOAA's Big Data Program (https://www.noaa.gov/information-technology/open-data-dissemination; webpage last accessed on 9 November 2022). The IASI level 1c data are available from EUMETSAT. The meteorological profiles used in this study are from ECMWF. The IASI $SO_2$ results for each orbit are available at https://catalogue.ceda.ac.uk/uuid/b80870de014a43a498fc2684e78f32af (webpage last accessed on 4 October 2023; Taylor and Grainger 2023). The data to reproduce Fig. 9 can be found in the supplementary material. The map animations for each eruptive event created with the ABI data can be accessed at https://ora.ox.ac.uk/objects/uuid:ca4e7a94-37c6-4d5f-b94a-1f287e661f8a (webpage last accessed on 2 October 2023; Taylor et al. 2023).

*Author contributions.* The IASI $SO_2$ retrievals and ash linear retrieval were run by IAT with guidance from RGG. The ABI images were studied by both IAT, RGG and SRP. ATP and SRP provided guidance on using the ABI data. IAT obtained height information from the ABI data, did the flux analysis and produced the figures. TAM and DMP provided insights into the volcanological processes and historical context of this eruption. All authors contributed to discussion of results and editing of the manuscript.

*Competing interests.* There are no competing interests

*Acknowledgements.* We are grateful to EUMETSAT for the provision of the IASI data, and to ECMWF for the meteorological data used. These datasets were accessed at the Centre for Environmental Data Analysis (CEDA) (EUMETSAT , 2009, 2014, 2021; European Centre for Medium-Range Weather Forecasts, 2012, 2021). The JASMIN data analysis facility was used to run the IASI retrievals and to run the code estimating the heights from ABI. We also acknowledge the NOAA big data programme for making the GOES ABI data available.

I.A.T., R.G.G., T.A.M. and D.M.P. acknowledge the support of the NERC Centre for the Observation and Modelling of Earthquakes, Volcanoes and Tectonics (COMET), a partnership between UK Universities and the British Geological Survey. This study was in part funded through NERC's support of the National Centre for Earth Observation (NCEO). I.A.T., R.G.G. and T.A.M. were supported by the Natural Environment Research Council (NERC) project VPLUS (NE/S004025/1). A.T.P. and R.G.G. were supported by the NERC project R4-Ash (NE/S003843/1). S.R.P.'s work on this study was funded as part of NERC's support of the National Centre for Earth Observation, award

ref. NE/R016518/1; and by a NERC Innovation fellowship, award ref. NE/R013144/1. D.M.P. acknowledges support from NERC Urgency grant NE/W000725/1. D.M.P. thanks Dan Goss, Jenni Barclay and Pat Joseph for discussions. I.A.T. thanks Anu Dudhia, Elisa Carboni and Richard Siddans for some of the code used. We would like to thank two anonymous reviewers for their constructive feedback which greatly improved the paper. Additionally, we would like to thank Steve Sparks for his feedback.

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
