# Peer review of "A satellite chronology of plumes from the April 2021 eruption of La Soufrière, St Vincent"

_Atmospheric Chemistry and Physics, 2022_

## Author Response (AR1)

**Peer review: Satellite measurements of plumes from the 2021 eruption of La Soufrière, St Vincent**

We would like to start by thanking both referees for their thoughtful reviews and constructive comments which have greatly helped to improve the manuscript. Before responding to the referee's comments, we would first like to acknowledge that we received feedback from discussions outside of the official peer review and these have led to some additional improvements. Also, our work on other projects identified some improvements to our IASI post-processing which we then applied to our results here. We will discuss these changes first and then go on to address the reviewers' comments. We hope that the referees agree that these help to improve the manuscript.

**Changes to IASI post-processing**

Additional studies we've been conducting in parallel to this one have identified some areas for improvement in our IASI post-processing. These have been applied to our La Soufrière results, leading to some small changes to the results, although not sufficient to change the manuscript's narrative. The post-processing changes will now be discussed followed by a short outline of how they have changed the results.

- **Ascending/Descending Artefact**

    As is typical in satellite remote sensing studies, we divided our results into descending (~9.30 AM local time at the equator) and ascending (~9.30 PM) tracks which splits the results in two with each being a compilation of tracks over a 24 hour period.  Following work on another project, it became evident that this can produce a discontinuity in the results where the day changes. The location of this discontinuity can lead to artefacts in the results which are not entirely avoidable, but which can be minimised.

    For IASI this discontinuity occurs at around 130 °E in the descending track and 60 °W in the ascending track (note this is very approximate based on visual observation on 10th April 2021). The discontinuity in the ascending orbits is in a bad position for the current study: occurring very close to the volcano (61.18 °W). The effect of this is not easily identifiable in the original set of results because the plume was travelling towards the east, which meant that part of the plume was being classified as ascending orbits for the following day.
    Having identified this, we have devised a solution which minimises this effect. The results are still divided into descending and ascending, but the time range has now been adjusted:
    - Descending: all descending tracks for orbits with measurement times starting between 0:00 and 23:59 on the day in question. This is the same as in our previous version
    - Ascending: all ascending tracks for orbits between 12:00 on the day in question, and 12:00 on the following day.
    By adjusting the time range for the ascending orbits we move the discontinuity to the other side of the globe which minimises the effect. We have added a paragraph explaining this to section 2.1.2:

    *'The results have been divided into descending (satellite travelling N to S; ~9:30 a.m. local overpass time at the equator) and ascending (satellite travelling S to N; ~9:30 p.m. local overpass time at the equator). This approach can lead to artefacts in the results at the point where the data crosses into the next day. This is particularly notable in the ascending orbits at La Soufrière, with the date change occurring around the location of the volcano. To minimise this impact, the descending and ascending results have been offset from each other: the descending results are a 24 hour composite of the descending nodes of orbits starting on each date, while the ascending results are compiled from 24 hours of ascending nodes of orbits starting from midday on each date and up until midday of the following day.  In this way the artefact is moved to the other*

*side of the globe, so minimising its impact. In this paper, the ascending results are referred to by the start date.'*

Similarly, we have mentioned how the data is divided in a number of the figure/table captions.

The impact of this change is most notable on 9 April 2021 which can be seen figure below. In this, a more substantial plume can be seen in the revised version.

[Figure]

**Figure:** Before (1st plot) and after (2nd plot) change made to method splitting results into descending and ascending. Example for 9th April 2021 (ascending node).

- **Gridding Change**

In another study we are conducting we noted that the gridding method was not optimal in all cases (having previously been optimised for another eruption). In particular, it was noted that it was it was not sufficiently filling gaps occurring where the iterative retrieval did not pass the quality control or gaps in the IASI field of view (the latter is not too important for La Soufrière as we are using all three IASI instruments). Following this we made a small edit to the gridding process, which has then been reapplied to the La Soufrière data. This has had a minimal impact on the results.

- **Tropopause height calculation**
Our previous tropopause calculation was based on the WMO definition of the tropopause:
*'that the tropopause occurs at the lowest level where the temperature lapse rate falls beneath 2 K km$^{-1}$ for at least 2 km'*
However, the shape of the some of the ECMWF temperature profiles meant that the tropopause height was often placed too low (based on visual inspection) as was noted in the previous manuscript. We've made a modification to the tropopause calculation. Now, in addition to the WMO definition, the code now requires the average temperature in the 10 km above the tropopause to be greater than the tropopause temperature. This has pushed the tropopause height up (it is now around ~17 km on average). This has led to a larger amount of $SO_2$ being classified as tropospheric (as we suggested would be the case in the previous version of the manuscript).

**Some of the key changes to the results following all the changes to the post processing described above:**
- Slight change to the lifetime estimate: previously 7.09±5.70 days, now 6.07±4.74 days
- The total $SO_2$ emission is now estimated to be 0.63±0.5 Tg compared to 0.57± 0.44 Tg.
- More $SO_2$ is now classified as tropospheric

The new version of figure 9 is displayed below:

[Figure]

**Figure:** Updated version of Fig. 9. (a) $SO_2$ distribution with height; (b) $SO_2$ distribution with latitude; (c) Total mass loading of $SO_2$ over time divided into tropospheric and stratospheric and indicating where this is uncertain; (d) Timeseries of the $SO_2$ flux.

Note: a mistake was noted in previous version of this figure – the incorrect colourbar range has been used for panel b. This has been corrected.

We've updated the manuscript with the new results. While these post processing changes have led to some changes to the results these are fairly minimal and do not alter the narrative of the manuscript.

**Changes to ABI results**

During the review process, we were made aware of another paper looking at this eruption using seismic data alongside the ABI data (Sparks *et al*., in press). This paper identified one event which we missed (starting at 12/04/2021 20:53 based on the seismic data). We examined the ABI data again and identified an emission at 21:13 UTC.

Re-examining all the ABI images again we revised some of the times. Smaller changes were made to the start times (up to 20 mins but more often 1-2 minutes) but there were more significant changes to the end times which were previously mentioned in the manuscript as being difficult to determine. The end times will be discussed further, later in this section. We also identified a possible further event at 20:16 on 12/04/2021 (a very small emission). We also decided to include the emission at 00:53 on 14/04/2021 as a separate event (previously this was mentioned in the comments but had not been listed separately). This takes the total number of events to 35. The results and manuscript have been updated accordingly. The changes in time led to slight variations in the heights obtained but not sufficient to change the narrative of the manuscript.

We've added a paragraph to section 3.1 explaining why there may be differences between Spark et al. (in press), Horvath et al. (2022) and this paper:

*'In contrast to this study, Horvath et al. (2022) count 49 explosive events between 9 and 22 April using the ABI data. The higher number reported by Horvath et al. (2022) is partly because they have divided event 2 (as termed in this paper) into multiple explosive events. Sparks et al. (in press) also document the sequence of eruptive events occurring during the April 2021 eruption. Their analysis is primarily based on seismic data, but they also make note of the first observation times with the ABI instrument. As expected, there are differences between the start times identified with the seismic and ABI instruments, as there is some time before the plume becomes visible with the satellite instrument. On the whole, their ABI observation times (only start time reported) agree well with the ones presented here, however there are a few differences. Firstly, there are differences in the timings of the events: they used the 10-minute full disc rather than the meso-scale data, and other differences may arise from the subjective nature of this exercise. Secondly, this study identifies some events which are not included in Sparks et al. (in press) timeseries, while Sparks et al. (in press), using the seismic data, is able to split event 2 (as referred to here) into multiple events. Finally, Sparks et al. (in press) also identify four phases to the eruption, and while there is some overlap with the phases outlined here, there is disagreement with the timings: reflecting the different datasets and metrics used.'*

We added a line in section 2.2.2 to clarify that the timings obtained with the ABI instrument are different to those observed using ground-based observations:

*'It should be noted that the start and end times reported here will be different to those observed on the ground. Here the start time is the first time that the plume is observed in the ABI data, while the end time incorporates the time taken for the plume to rise and disperse away from the volcano. For this reason, the times reported here are different to those reported in Sparks et al. (in press) using seismic data.'*

A further change made was to better represent the uncertainty of the end times. We had previously noted that this was particularly challenging to represent. To highlight this uncertainty, we now present the end times as a range for most of the events.

**Additional changes:**

- The ABI plots have been tidied. In particular, the true colour images now show white during night-time.
- When computing the total mass emitted we no longer discount the negative fluxes
- We've adjusted some of the plots to try improve their accessibility to those who are colour blind.
- We added references to other papers looking at La Soufrière (e.g. Koukouli *et al.* 2022; Sparks *et al.* in press; Esse *et al.* 2023; Bruckert et al. 2023). We also added a reference to the Geological Society special issue (April 2024) - for which some papers are already available online. We expanded a little on the similarities and differences with Horvath *et al.* (2022). And in the results section include more comparisons with other papers.
- We identified that there was an eruption at Sangay on 12 April 2022. This plume combines with that from La Soufrière and the two can be difficult to distinguish. This may contribute to the total masses recorded by IASI. We have made a note of this eruption in sections 3.2 and 3.3:

  *'Note that there is an emission of $SO_2$ from an eruption at Sangay in Ecuador from 12 April (GVP, 2021d). This combines with the plume from La Soufrière and the two cannot be easily distinguished from each other from 13 April.'*

  *'A total mass timeseries, derived from the IASI iterative retrieval output for the -45 to 45° N and -180 and 180° E region, is shown in Fig.9c. Note that other volcanoes (e.g. Sabancaya in Peru; Sangay in Ecuador) were erupting at this time and any $SO_2$ from these eruptions entering the region will affect the total mass, e-folding and flux estimates.'*

- We've made a note that underestimations in the total masses will affect the flux results in section 3.3.
- We expanded the number of days and orbits that the linear ash retrieval was run for so we have expanded Table 1.
- We've added a line to illustrate how the IASI $SO_2$ error masses are calculated and why our errors are so large:
  *'The total mass errors are computed in the same way which may lead to an overestimation of the mass error, but to sum the errors in quadrature could lead to an underestimation due to the systematic errors. Note that this overestimation in the errors is carried forward into the flux and total emission errors.'*
- We've made some changes to improve clarity, expand on some points or fix small errors in the text.

**Responses to Referee #1**

We thank referee #1 for their very kind and helpful comments.

**1) L36: Is there an acronym for SMS-1?**

We have updated this line to read:

*'Observations from the infra-red radiometer on the Synchronous Meteorological Satellite - 1 (SMS-1) were used... '*

**2) L55-56: maybe a mention to the temporal resolution of satellite observations is needed here, in particular for the high-temporal resolution of geostationary instruments.**

We've added:

*'Geostationary instruments with a high temporal resolution (e.g. up to 30 seconds) are extremely valuable for identifying hazardous plumes and can also help to characterise eruptive events (e.g. Gupta et al. 2022; Prata et al. 2022).'*

**3) L99: RTTOV must be introduced and defined earlier in the text**

To introduce RTTOV we added a line to the paragraph before (first paragraph of section 2.1.2):

*'In broad terms, this method works by comparing the IASI measured spectra against $SO_2$ spectra simulated by the fast-radiative transfer model RTTOV (version 9; Saunders, 1999), see Carboni et al. (2012, 2016, 2019) for more details'.*

**4) L102-103: "column amounts exceeding 0 DU", you mean "positive column amounts"? And also "positive heights"?**

As requested by referee 2 this paragraph has been removed.

**5) L122-123: not clear what do you mean here "Note that...analysis."**

This has been rewritten as:

*'Note that there are spectral similarities between volcanic ash, desert dust and desert surfaces which can lead to false ash flags in desert regions such as the Sahara (e.g. Prata et al. 2001; Simpson et al. 2003; Park et al. 2014). To avoid this, the ash linear retrieval has been run for a smaller region (-5° to 25° N, -68 to -20° E) than the main analysis'*

This has been moved to the paragraph where the ash linear retrieval is introduced to improve clarity.

**6) Eq. 1: why not putting this in its integral shape, i.e. as an exponential decrease? And why not using another symbol for the e-folding time - lambda may be confusing in this context, and taken as a wavelength - ?**

We feel that the equation in its current form clearly conveys how it has been used in this study. As suggested by referee #2 are now using τ for e-folding time.

**7) Is Fig. 2 more pertinent in the "Results" section than here in the "Data" section?**

We prefer to keep Fig. 2 in the section 2.1.3 as we have used this to calculate the a priori lifetime used in the optimal estimation approach and it is not the final lifetime estimate presented in the results section.

**8) Maybe this information might be more visible in a Table.**

This refers to the what the different colours in the false colour images refer to.

This has been added:

Table 2. Outline of how to interpret the ABI false colour images (see GOES-R 2018 for more information). An example image is shown in Fig. 3

| Colours | Explanation |
|---|---|
| Brown/Orange | Optically thick cloud (ice, ash or meteorological cloud) |
| Red/Pink | Less optically thick volcanic ash |
| Bright green/blue | $SO_2$ |

This was a helpful suggestion and following this comment, we also decided to represent the limitations of using BT measurements to obtain height estimates within a table.

**9) L259-260: please add references for the possible reasons of differences between ground based and satellite observations.**

We've expanded on this paragraph and added references:

*'However, differences between ground based and satellite measurements can arise for a number of reasons including the different viewing angles and assumptions, poor visibility on the ground, the plume rising too high for the height to be accurately obtained from the ground and different definitions of plume height (e.g. Tupper et al. 2004; Tupper and Kinoshita, 2003; Tupper and Wunderman, 2009).'*

**10) Fig. 1, 7 and 8: please consider to annotate the ash and SO2 amounts color bars in linear scales and not log10**

We've updated the plots to have colour bars in linear scales rather than $log_{10}$. We've added a gray background to the $SO_2$ maps in Fig. 7 (and in the supplementary information) to make it easier to identify the location of the plume when the column amounts are low. We had to significantly restrict the colourbar range for Figs. 1 and 7 to better show the lower amounts of ash/$SO_2$.

**11) As the emitted SO2 mass for the recent Hunga Tonga-Hunga Hapa'ai eruption was initially estimated at 0.4 Tg (https://www.frontiersin.org/articles/10.3389/feart.2022.976962/full), this is maybe an interesting event to compare with your mass estimations for La Soufrière. Please note that the impact of the Hunga Tonga-Hunga Hapa'ai eruption on the stratospheric aerosol layer and climate was much larger than what expected for such a relatively small estimations of the SO2 emissions (https://www.nature.com/articles/s43247-022-00618-z). This discrepancy of estimated emissions and the impacts is also pertinent in your discussion, in my opinion.**

It is not clear how a comparison with Hunga Tonga-Hunga Hapa'ai (HT-HH) adds to the paper. Although the emitted $SO_2$ amounts were similar the principal climatic effect of HT-HH was owing to the enormous amount of water vapor injected into the stratosphere by HTHH (Jenkins et al, 2023; 10.1038/s41558-022-01568-2).

We agree that the Hunga Tonga-Hunga Hapa'ai (HT-HH) eruption is of great interest in terms of the eruption and plume dynamics and its climate impact. However, we have been careful to avoid comparing our IASI results from La Soufrière with those from other instruments/techniques for other eruptions due to significant differences that can arise (particularly between UV and infrared instruments). Instead we've compared against eruptions studied by Carboni *et al.* (2016) to ensure we are comparing against a consistent method.

We are hoping to publish our results on the HT-HH eruption in the near future which may be a more appropriate place to compare the two eruptions, and the results from different instruments.

**12) L320-321: with which satellite instrument the SO2 mass of 0.4 Tg was estimated by the GVP?**

Unfortunately the GVP report referenced here does not specify which instrument was used to get the 0.4 Tg $SO_2$ mass.

**Responses to Referee #2**

We would like to thank referee #2 for their very thorough review which has helped to improve the manuscript.

We are highlighting here some of the key parts in the summary section of the review before going on to discuss the specific comments.

**The text within sections 3.2 and 3.3 remained at rather a basic level, and needs to be improved to highlight the atmospheric processes involve before the manuscript is ready for publication. There needs also to be a strengthening of the overarching science narrative, and for this to be reflected in a revised title, over and above the basic "analysis of satellite measurements for the La Soufrière eruption".**

We have updated the title to:

*'A satellite chronology of plumes from the April 2021 eruption of La Soufrière, St Vincent'*

**Considering the science interpretation in section 3.2, the specific revisions here are to better identify the relevance of the sudden change in wind regime seen in Figure 4a, re: some parts of the plume being dispersed to the west (easterly) whilst the majority of the plume is dispersed towards the east (westerly). Figure 4a very nicely shows both the mean and standard deviation of the wind direction, then clearly identifying the 8-16km altitudes consistently had easterly winds, with westerly winds only occurring at higher altitudes, or near the surface. Figure 5 then represents a further analysis of the zonal and meridional wind components at three distinct time-periods, labelled eruptions 1, 2 and 3. The corresponding text in the submitted manuscript (lines 218 to 235) currently does not (yet) explain the significance of the variations mentioned, presenting only each variation in the Figure without a coherent narrative of the main findings here, in relation to the plume's dispersion from these 3 main explosive phases of the eruption. It may be that the text can only hint at the interpretation at this stage, and that the themes of the interpretation can then be picked up later within the SO2 analysis in Figures 7, 8 and 9, but the text here (lines 218 to 235) should already be introducing the clear vertical shift in wind regime revealing then an important variation relevant for the dispersion of the plume. Currently the text has only rather abstract terms such as "supporting a tropospheric solution", when it should make a clearer statement such as the westerly winds only being present in the upper altitudes or so.**

Following this comment, we have made a number of changes to different sections of the manuscript to highlight the information that can be gained from the wind directions.

In section 2.2.2 we've expanded the describing of the wind profiles:

*'Figure 4a shows the average and standard deviation of the wind direction with height and the average wind speed with height is shown in Fig. 4b. At heights of less than 5 km the wind is primarily travelling to the west (e.g. 267.5° at 4.3 km) with wind speeds of less than 10 m s$^{-1}$. At around 5 km the wind direction shifts to the east. Between 7 and 17 km the wind direction is fairly consistent (varying between 101 and 126°) before shifting back to the west at 19 to 20 km. The easterly winds in the stratosphere are a characteristic associated with a phase of the Quasi-Biennial Oscillation (QBO): alternating strong easterly and westerly zonal winds around the equator which propagate through the stratosphere to the tropopause (Reed et al. 1961, Baldwin et al, 2001).'*

We added the tropopause height to Fig. 4 a+b.

*In section 3.2, we've related the transport direction plume to the wind directions shown in Fig. 4a:*

*'... A stronger signal is then visible in the descending orbits on 10 April fanning out to the east of the volcano across the North Atlantic. The general east and south-eastward transport of the plume between 9 and 11 April implies that the bulk of $SO_2$ has been emitted into the troposphere, with Fig. 4a indicating wind directions between ~90-140° dominating in the troposphere between 8 and 17 km... On the 12 April, while the bulk of the plume is still advancing towards the east (Fig. 7), a fraction of the plume travels to the south and west of the volcano. The wind directions shown in Fig 4a, imply that for westward transport, either some $SO_2$ has been emitted into the lower parts of the troposphere (less than 5 km) or more likely that some $SO_2$ has been emitted, or has been lofted, into the stratosphere.'*

Prompted by this comment, we noted that there was perhaps not as much variability to the heights in the southern part of the plume as we would have expected and that is seen in Koukouli et al. (2022). We experimented with some different retrieval setups, varying the first guess of the plume, as the retrieval can sometimes get stuck at a local minimum. We did not see any change to the result. We've added some discussion of this to section 3.4:

*'There is no obvious gradient to the heights in the upper part of the plume as can be seen in IASI $SO_2$ height measurements shown in the supplementary material of Koukouli et al. (2022) for 10 and 11 April (using the Clarisse et al., 2014 method). Koukouli et al. (2022) show heights increasing from south to north of the plume: which matches well with the wind directions shown in Fig. 4a. The IASI retrieval used here relies on temperature and water variations in the atmosphere, which do not vary as significantly around the tropopause, which may affect the results. Multiple retrieval setups were explored (including varying the retrieval first guess height and varying the plume thickness) but the results were similar in each case. Nevertheless, there is a broad agreement with Koukouli et al. (2022) which reports average heights of 15.7 ± 1.16 km for IASI using the Clarisse et al. (2014) retrieval and 14.94 ± 3.87 km from TROPOMI (Hedelt et al. 2019 method) (note that these averages are based on a subset of the plumes). Additionally, there is agreement with the 13 to 15 km injection heights obtained by Esse et al. (2023) ...'*

**Similarly, the section 3.3 text only discusses the actual variations in the Figure without communicating the basic processes involved, in this case there the progressing oxidation of the volcanic SO2 during downstream transport of the plume, with any removal from the stratosphere on the timescales considered.**

We've added an additional line to this section which discusses the decrease in the total $SO_2$ mass and moved our discussion of the e-folding time up to this section:

*'From 13 April the total mass of $SO_2$ is shown to fall: as the $SO_2$ is removed from the atmosphere through deposition, by conversion to sulfate aerosol, or dilution below the detection limit of the instrument. The e-folding time used here (see section 2.1.3) describes this loss process. This varies with a number of factors including the latitude of the volcano, the injection height of the plume, meteorological conditions, cloud cover, water vapour, season and the presence of ash (Carn et al. 2016, Zhu et al. 2020, Schmidt et al. 2022, Zhu et al. 2022). Typically, the e-folding time varies from hours to days in the lower troposphere to weeks in the stratosphere. A first estimate of 5.47 days for the e-folding time was estimated by fitting Eq. 1 to the IASI total $SO_2$ masses between 23 to 30 April 2021. Following the application of the Carboni et al. (2019) method to compute the flux and average e-folding time, the average e-folding time was adjusted to 6.07±4.74 days. This is in line with other eruptions including Jebel at Tair (2007) and Merapi (2010), both volcanoes in the tropics and which emitted plumes between 15 and 18 km, and had e-folding times of between 2 and 4 days (Carn et al., 2016). Given that there was ash emission during the La Soufrière*

*eruption, it is possible that the e-folding time could have been reduced as a result of accelerated oxidation of $SO_2$ due to reactions on the ash surface, as was seen for the Kelut eruption in a study by Zhu et al. (2020). Some of the $SO_2$ emitted during the La Soufrière eruption was converted to sulfate aerosol, as is shown in Babu et al. (2022) and Bruckert et al. (2023).'*

We believe any further discussion of sulfate formation is beyond the scope of this study.

**In both cases (sections 3.2 and 3.3), there is a notable absence of citations of studies analysing these studies from other highly explosive eruptions reaching the stratosphere, with section 3.2 needing to mention also the well-known dominant mode of variability in the tropical stratosphere, the quasi-biennial oscillation (Reed et al., 1961; Baldwin et al., 2001) and it role in the dispersion of volcanic clouds in the stratosphere (e.g. Trepte and Hitchmann, 1992; Trepte et al., 1993; Langford et al., 1995).**

We found a more natural place to refer to the QBO in section 2.2.2 in the introduction to the wind profiles:

*'The easterly winds in the stratosphere are a characteristic associated with a phase of the Quasi-Biennial Oscillation (QBO): alternating strong easterly and westerly zonal winds around the equator which propagate through the stratosphere to the tropopause (Reed et al. 1961, Baldwin et al, 2001).'*

Any further discussion of the QBO is beyond the scope of the paper.

**Similarly, re: section 3.3., there should be citations of some studies that have analysed the daily progression of stratospheric SO2 burden within other tropical volcanic SO2 clouds, e.g. Guo et al. (2004) for Pinatubo, Zhu et al. (2020) for 2014 Kelud and Zhu et al. (2022) for 2022 Hunga-Tonga. There should also be mention of the recent recognition of the potential for ash particles to accelerate the SO2 conversion via heterogeneous oxidation on the surface of the ash particles (Zhu et al., 2020).**

We've generally increased the referencing throughout, in particular adding references to new studies looking at the La Soufrière eruption.

As outlined above, we moved our discussion of the e-folding time to section 3.3 and within this added further references to factors that influence it (Carn et al. 2016, Zhu et al. 2020, Schmidt et al. 2022, Zhu et al. 2022).

We do not believe that comparing against other eruption such as Pinatubo and Hunga Tonga-Hunga Ha'apai adds much to the manuscript as there are significant differences between these eruptions and their atmospheric impacts.

**The Abstract was rather poorly worded, and 9 of the first 10 of the specific revisions focus on improving the Abstract. I also found the sentence structure seemed to have very short sentences, then unsuited for the scientific interpretation required within an ACP article. Specifically, I wonder whether the autuor may have used a grammar-checker, and amended their original sentence structure into some recommended series of short sentences, which then no longer communicate the depth of explanation in the original sentence structure? In any case, I flag up here that several of the specific revisions are where the manuscript had very short sentence that didn't seem to communicate any content. And then I wonder if previously the wording did communicate a coherent structure, but this was lost in some shortening of sentences via an auomated grammar-checking software or similar?**

We would argue that the sentence structure is a matter of personal preference and that shorter sentences are often encouraged in scientific writing to avoid misinterpretation. Nevertheless, we have made a

number of changes, following your suggestions, and the abstract in particular has been significantly improved.

**1) Title -- I'm suggesting here the authors consider amending the title of the manuscript, with**

**"Satellite measurements of plumes from the 2021 eruption of La Soufrière" not really communicating the science focus of the study. I think the information in Tables 2 and 3 could potentially be called an "eruption chronology" or so, in relation to the timing of the individual explosive events the analysis identifies. I think including the duration of the series of explosive events explicitly (14 days) could aleo help the reader appreciate the multiple large-magnitude explosive events. And a specific suggestion is to change the title to "A 14-day explosive eruption chronology for the April 2021 eruption of La Soufrière, St Vincent". I leave it to the authors to decide whether to change the title, but some re-wording, certainly to include the month "April" with the suggest to also include some better specifics within the title.**

We have amended the title:

*'A satellite chronology of plumes from the April 2021 eruption of La Soufrière, St Vincent'*

**2) Abstract, line 3 -- Re-word "were observed by a multiple satellite instruments." Although the Abstract can include some introductory sentence, the current Abstract has the first two sentences as the context for the study, and suggest to merge the 2nd and 3rd sentences into 1 sentence focused on the specific science focus of the manuscript.**

**As per my general comments above, I think the analysis of the geostationary satellite measurements in relation to defining an eruption chronology represents a valuable and novel aspect of the study. My suggestion here would be to incorporate this specific outcome from the 1st part of the study into a revised wording of the current 2nd and 3rd sentences. Maybe have the new 2nd sentence begin "This study analyses geostationery sallite measurements for a 14-day chronology of the series of explosive eruptions of the La Soufrière volcano in April 2021." or similar. To save words, the location of La Soufrière (St Vincent in the Eastern Caribbean) does not need to be communicated in the Abstract, with this given within the manuscript text. With then the 3rd sentence focusing on the way the study combines these then with SO2 measurements from the IASI sensor on the polar-orbiting MetOp satellites.**

**Within this revised 3rd sentence, please re-phrase "looks at these plumes" to more scientific wording.**

**I'd advise to keep the acronyms as acronyms here, with these being fine to be explained in the main text of the article. It's currently very clunky with multiple brackets within this early part of the Abstract.**

We found it difficult to rewrite these lines exactly as suggested. However, we have made a number of adjustments following your recommendations including reducing the context to the first two sentences:

*'Satellite instruments play a valuable role in detecting, monitoring and characterising emissions of ash and gas into the atmosphere during volcanic eruptions. This study uses two satellite instruments, the Infrared Atmospheric Sounding Interferometer (IASI) and the Advanced Baseline Imager (ABI), to examine the plumes of ash and sulfur dioxide ($SO_2$) from the April 2021 eruption of La Soufrière, St Vincent. The frequent ABI data has been used to construct a 14-day chronology of a series of explosive events at La Soufrière; which is then complemented by measurements of $SO_2$ from IASI which is able to track the plume as it is transported around the tropics.'*

The ACP guidelines state that acronyms should be defined when first introduced (https://www.atmospheric-chemistry-and-physics.net/submission.html). We have removed the satellite names to reduce the number of acronyms in the sentence.

We feel it is important to mention that the volcano is on St Vincent as there is another Soufrière volcano on Guadeloupe.

**3) Abstract, lines 5-7 --> Re-word this sentence to have the main "object" of the sentence at the start of the sentence, this being the 32 eruptive events that are defined, with the methodological aspects coming after the specifics -- it just makes the sentence easier to read.**

This has been reworded as:

*'A minimum of 35 eruptive events were identified using true, false and brightness temperature difference maps produced with the ABI data.'*

**Also, suggest to change the word "character" instead to "characteristics", as I think this is what is meant, in relation to this referring to the content within the "Notes" column of Table 2.**

Rewritten as:

*'The high temporal resolution images were used to identify the approximate start and end times, as well as the duration and characteristics of each event.'*

**4) Abstract, line 7 --> Change "The ABI images were used..." instead to "The high temporal resolution BTD images from the geostationary ABI measurements were used..". Suggest to add "in this 14 day period" after "32 eruptive events" and replace "eruptive events" with "large-magnitude explosive events". (The acronym BTD can be introduced in the preceding sentence.)**

We included 'high temporal resolution':

*'The high temporal resolution images …'*

We did not specify BTD images as these were used alongside the true/false colour images. We've not added 14 day period to this sentence as we used '*14-day chronology*' in the previous one.

We are still using the 'eruptive event' as not all the events studied are 'large magnitude': particularly the events in the waning stage.

**5) Abstract, line 8 --> Re-word "In this way the eruption has been divided into four phases..." instead to "From this analysis, we define four distinct phases of the 14-day eruption", each consisting of multiple events" or similar. ("In this way" is not scientific enough and good to be clear this is within the 14 days (9th to 22nd April). Also good to introduce the terms "events" and "phases").**

Rewritten as:

*'From this analysis, four distinct phases within the 14-day eruption have been defined, each consisting of multiple explosive events with similar characteristics: …'*

**6) Abstract, lines 9-12 -->** There are 3 short sentences here which would seem better re-worded into a single sentence, the 2nd sentence clearly continuing from the content communicated in the first very short sentence, and the 3rd very short noting the altitude range. As per my general comments above, this is an example where the writing in very short sentences seems to me to lose the ability to communicate the scientifically-related aspects. I wonder if this change is a result of using a grammar checker that has advised to split the text in this way? The current 1st sentence communicates only methodological aspects and the order in which these 3 aspects are communicated in a revised sentence should also be changed. In my opinion it is best to begin with the main object of the sentence rather than the methodological aspects. y specific sugestion then is to have a merged sentences communicating all 3 aspects within a wording such as

**"Analysis of the IASI SO2 measurements shows the dispersing La Soufrière SO2 cloud had a highly complex structure, the multiple explosive events generating several plume enhancements in the altitude range 13 to 19km", or something similar.**

We have combined the first two sentences:

*'It is likely that the multiple explosive events during the April 2021 eruption contributed to a highly complex plume structure which can be seen in the IASI measurements of the $SO_2$ column amounts and heights.'*

We have kept the height results as a separate sentence (now beneath a discussion of the wind directions and height information that this gives us) but combined them with the following line (on troposphere/stratosphere):

*'The retrieved $SO_2$ heights show that the $SO_2$ was largely concentrated between 13 and 19 km, with the majority of the $SO_2$ being located in the upper troposphere and around the height of the tropopause, and with some emission into the stratosphere.'*

**7) Abstract, lines 12-14 -->** Again there are 2 sentences here in strangely worded very-short sentences. The wording seems to be compartmentalising each individual aspect into a separate sentence, with no over-arching structure to communicate the science behind these related aspects. As per my general comments above, a key aspect here is to mention the fact that whereas the majority of the plume dispersed eastwards, part of the plume dispersed towards the west, and this bi-directional plume transport must be reflecting the wind structure shown in Figure 4a. Depending on how the authors re-word the section 3.2 content, the main finding in that regard should be stated within this re-wording of these 2 sentences of the Abstract. I'd suggest to potentially word as **"We show that most of the SO2 was transported eastwards (westerly flow),but a proportion of the plume was transported westward (easterly), reflecting the higher-altitude wind regime in the lower stratosphere (or similar).** This comment relates to the specific comments xx and yy to improve the interpretation within the section 3.2 results text.

We edited this section to:

*'The bulk of the $SO_2$ from the first three phases of the eruption was transported eastwards, which based on the wind direction at the volcano, implies the $SO_2$ was largely in the upper troposphere. Some of the $SO_2$ was carried to the south and west of the volcano, suggesting a smaller emission of the gas into the stratosphere: there being a shift in wind direction around the height of the tropopause.'*

**8) Abstract, lines 14-16 -->** Yet again, there are 3 sentences here, that should be joined together into one sentence, to give the information in the 2nd and 3rd sentence as a continuation of the same sentence. It makes no sense to separate this information in this way. In particular the middle sentence reports very little information, and the words here should refer to the temporally varying emissions flux (aligned to the main

**"emission chronology" narrative for the manuscript. Suggest to continue with the 2nd sentence wording changed to ", with derived emissions fluxes highest on 10th April, phase 3 of the eruption..." with then the last part, "... with later explosive events of leser magnitude and injection height, and generally decreasing". Or similar wording to this.**

This has been rephrased:

*'Using the IASI SO₂ measurements, a timeseries of the total SO₂ mass loading was produced, with this peaking on 13 April (descending orbits) at 0.31±0.09 Tg. Converting these mass values to a temporally varying SO₂ flux demonstrated that the greatest emission occurred on 10 April with that measurement incorporating the second phase of the eruption (sustained emission), and the beginning of the pulsatory phase. The SO₂ flux is then shown to fall during the later stages of the eruption: suggesting a reduction in eruptive energy, something also reflected in height estimated obtained with the ABI instrument.'*

**9) Abstract, lines 16-18 --> The wording should also be improved here, with the start of the sentence again better communicating the main information rather than the method. The current wording "By summing the IASI SO2 flux results, it is estimated that" should be reduced, to "We derive a total emissions flux for the 0.57 +/- 0.44 Tg of SO2, ...." The reader will be aware this is derived from the IASI SO2, it doesn't need to be stated again, and only serves to detracts from communicating the main result to again state the specific sensor analysed.**

Rephrased to:

*'A total SO₂ emission of 0.63±0.5 Tg of SO₂ has been derived.'*

**10) Abstract, lines 19-22 -- This is a nice last part of the Abstract to compare back to the 1979 eruption, but again the wording of this last part of the Abstract would be better in a flowing sentence than inthe compartmentalised into 3 short sentences. Suggest also to re-word "be prepared for future activity" more specific wording.**

We've combined the first two sentences here:

*'There are a number of similarities between the 1979 and 2021 eruptions at La Soufrière, with both eruptions consisting of a series of explosive events with varied heights and including some emission into the stratosphere.'*

Final sentence has been rephrased as:

*'These similarities highlight the importance of in-depth studies into these eruptions, and the valuable contribution of satellite data, as such studies help to learn about a volcano's behaviour, which may help to be better prepared for future eruptive activity.'*
* * *
The full abstract now reads:

[revised manuscript text omitted]

"la" in french is the definite article "the" in English. 'La Soufrière' translates from French to "The Sulfur Source". It's a tautology to use two definite articles.

The line now reads:

*'La Soufrière (61.18° W, 13.33° N, summit elevation of 1220 m), a volcano on the island of St Vincent ...'*

**and replace "located" with a comma, inserting also a comma after "the Caribbean".**

Sentence rephrased so this no longer applies.

**Suggest to delete "including the slow extrusion of a lava dome", to instead stay focused on the science topic of the manuscript,**

We believe this line provides important context about activity before the explosive eruption in April. It is likely that the dome contributed to pressurisation of the system and was destroyed in the first explosive event and is therefore relevant to the science topic of the manuscript.

**and re-word "having been in a lower level state of eruption" instead to "having been a series of low-explosivity events since late December" or similar.**

We would like to keep this as it is. Activity before the April 2021 events did not consist of explosive activity but instead effusive activity.

**The "in the Caribbean" could be deleted since this information is given in the sentence after.**

Done

**12) Introduction, line 38 -- change "detected" to "measured" (the SAGE instrument did more than detect the enhancement, it measured the magnitude of the enhancements). Also change "following" to "from".**

To avoid repetition of 'measure' in the same sentence:

' the Stratospheric Aerosol and Gas Experiment (SAGE) measured stratospheric aerosol from the eruption'

**Similarly, change "were detected by" to "were observed by".**

Done

**13) Introduction, lines 49 to 51 -- The word "satellite is used twice in this sentence, and suggest to change the first instance from "Advances in satellite technology" instead to "remote sensing technology", also reducing the current text "in the last four decades mean that in 2021 multiple aspects" instead to "in the four decades since the 1979 eruption, multiple aspects" and changing "the volcano's activity" instead to "the 2021 activity". Replacing "were studied with satellite instruments" to "were monitored from multiple satellite instruments".**

We've rephrased this as:

'Technological developments in the four decades since the 1979 eruption mean that numerous aspects of the 2021 eruption were observed with multiple satellite instruments, including thermal anomalies, dome growth, lightning and the evolution of the $SO_2$, ash and sulfate plumes… '

**14) Introduction, line 53 to 56 -- Again the split into two sentences detracts from the science communication here. Suggest to improve the wording by changing "plume properties. This is essential for assessing the potential hazard of volcanic plumes to aircraft" instead to "plume properties, essential for assessing potential hazard to aviation",**

This has been changed to:

*'Measurements made by satellite instruments allow the detection of SO$_2$ and ash, and quantification of the plume properties, which is essential for assessing the potential hazard to aviation'*

**and also change "providing estimates of the eruption parameters" to "enabling to provide more sophisticated information on eruption source parameters". Please add "Marshall et al. (2018)" in addition to Aubry et al. (2021) re: eruption source parameters.**

Updated to:

*'... providing more sophisticated information on eruption source parameters (Aubry et al. 2021).*

*While an interesting paper, Marshall et al. (2018) does not discuss the use of satellite data to determine eruption source parameters.*

**15) Introduction lines 57 -- state which (or how many -- 2 or 3?) of the MetOp satellites (-A, -B or -C) the IASI data included in the analysis. It's good that the word "satellites" is plural but, also since the better to state specifically the number of these -- two or three? And maybe keep the text general here, replacing "on the MetOp satellites" to "on two polar orbiting satellites".**

The text has been updated to:

*'on the three MetOp satellites'*

Additionally, we have added a line to section 2.1.1 to be clearer about what we used in this study:

*'MetOp-A, -B and -C launched in 2006, 2012 and 2018 respectively, with data from all three instruments used in this study'*

**16) Introduction line 58 -- change "to study the" to "to analyse the" -- the paper is presenting an analysis of the measurements, more than simply studying the eruption.**

Done

**17) Introduction line 68 -- insert comma between "pixels" and "each with" and delete "at nadir" after "square".**

We've added the comma. We've not deleted the 'at nadir' as it is important to understand that these dimensions only hold in these conditions.

**18) Section 2.1.1, lines 75-78 -- These 2 sentences should be merged to just 1, and the wording reduced, with the words "have been developed to obtain information about other volcanic gas species including" changed to "have been developed also for". Changing ". Additional work has been done on the retrieval of sulfate" instead to worded ", and for sulfate", then enabling this to continue in the same sentence.**

This has been rephrased as:

*'Retrieval techniques have also been developed for other volcanic gases and aerosols including H$_2$S (Clarisse et al. 2011), CO (Martinez et al. 2012), sulfate (Guermazi et al. 2021) and ash particles...'*

**Insert "the infra-red method means that" before "there is no break in coverage", and then reduce "associated with the loss of solar radiation at night and during high-latitude winters" instead to "at night or during high-latitude winters".**

This has been rephrased as:

*'Each IASI instrument obtains near-global coverage twice a day, and being infrared measurements means that there is no break in coverage at night and during high latitude winters'*

**19) Section 2.1.1, line 79 -- Hyphenate "near global" to "near-global" and add comma after "twice a day".**

Done

**20) Section 2.1.2, lines 80-81 -- I'd suggest to delete this sentence but if the authors prefer this to remain that is OK.**

We would prefer to keep this sentence to emphasise the value of using IASI for studying volcanic plumes

**21) Section 2, line 85 -- Change "two methods have been employed for studying the" instead to "two methods have been used to analyse" -- again, the methods are doing more than "studying" the SO2 plumes, and the word "employed" seems strange in this contex.**

Done

**22) Section 2.1.2, line 86-87 -- Insert "column" after "elevated quantities of" --> it's the elevation of the column SO2 the method detects (at least to my understanding), and that's an important caveat to communicate there.**

Changed to:

*'…which contain elevated quantities of columnar $SO_2$'*

**The very short sentence beginning "Full details of" should be included simply as a continuation of that sentence, changing ". Full details of this method can be found in Walker et al. (2011, 2012)." instead to ", see Walker et al. (2011, 2012)."**

Done

**23) Section 2.1.2, line 88 -- Improve this sentence by deleting "which is able to", then changing "quantify information about the plume, including..." instead to ", quantifying additional information about the plume, including..."**

See response to comment 24

**24) Section 2.1.2, lines 89-90 -- Change "In this study it has been applied.." (the first sentence of this paragraph already starts "In this study", and better to re-word to "The method is applied.." Again, merge these two very short sentences into 1 sentence, changing ". The retrieval has been run in much the say way as is described in Carboni (2012, 2016, 2019)" instead to ", (see Carboni et al., 2012, 2016, 2019)." Since the differences are**

explained in the text, there is no need to state "in much the same way as" -- the reader may initially think it's the same method, but will read the next sentence, there communicating the slight differences in the methodology (without that needing to be stated explicitly).

We've chosen to address comments 23 and 24 together as we made further changes to this paragraph as referee #1 requested that we introduce RTTOV (see comment 3 from referee #1). We've taken onboard the comments made here and hope the paragraph now reads better and is more concise:

*'In this study, two methods have been used to analyse $SO_2$ plumes from the La Soufrière eruption. The first method is a linear retrieval which is applied in this case to detect pixels which contain elevated quantities of columnar $SO_2$, see Walker et al. (2011, 2012). The second method is an optimal estimation retrieval scheme, which has been applied to the flagged pixels to quantify the column amount, height and the effective radiating temperature, and the errors associated with each of these. In broad terms, this method works by comparing the IASI measured spectra against $SO_2$ spectra simulated by the fast-radiative transfer model RTTOV (version 9; Saunders, 1999), see Carboni et al. (2012, 2016, 2019) for more details. A few changes have been made to the retrieval setup:'*

**25) Section 2.1.2, line 99 -- Change "of the RTTOV (the forward model used) pressure levels" to "of the pressure levels in the RTTOV forward model".**

Done

**26) Section 2.1.2, lines 100-101 -- Delete this sentence -- it refers to future work not applied in this study.**

Done

**27) Section 2.1.2, lines 102-104 -- This sentence concerns information too specific to the coding implementation of the algorithm. Whilst in an ESSD or GMD manuscript this level of information might be able to be incorporated, for an ACP manuscript this is too specific to the implementation of the methods. Please delete as it does not provide information on the scientific methods, only the implementation of these at the coding level.**

Done

**28) Section 2.1.2, lines 111-112 -- Again there are two very short sentences here, which seem better as one long-ish sentence. Please change "the iterative SO2 retrieval. This analysis demonstrated that ash with..." instead to "the iterative SO2 retrieval, demonstrating that ash with...".**

Done

**Also change "can significantly effect" to "can significantly affect".**

Done

**29) Section 2.1.2, line 116 -- "This retrieval is run at three pressure levels (400, 600 and 800 hPa) to obtain three estimates of the optical depth". This sentence needs to be re-worded, or explained by the authors. A reader will assume the "optical depth" must be for the column, rather than specific to any given altitude, and**

**this wording here is confusing. If the method is analysing three different threshold altitudes (e.g. for altitudes above some minimum-altitude pressure level) then change "at three pressure levels" instead using three different threshold pressure levels" to make that clear. But if it is actually assessing an optical depth only for a particular shallow-layer of the atmosphere at these different pressures, then the wording "optical depth" needs to be changed to "aerosol extinction" or "optical depth contribution" or similar (to ensure the reader is aware it's chcking for ash at these levels.**

The linear retrieval is run assuming the ash layer is at one of three different pressure levels (400, 600 and 800 hPa) and it obtains an estimate of the ash optical depth. We have rephrased this line to improve clarity:

*'This retrieval is run at three pressure levels (400, 600 and 800 hPa) to obtain three estimates of the ash optical depth. Pixels are then flagged as containing volcanic ash if any one of these ash optical depths exceeds a threshold.'*

**Please always include "ash" wherever "optical depth" is stated here (assuming I'm correct that this is the optical depth of the ash particles --> i.e. "ash optical depth" or "ash optical depth contribution".**

Done. With the exception of *'… demonstrating that ash with an optical depth'* where the inclusion of ash after makes this clear.

**30) Section 2.1.2, Caption to Figure 1 -- My understanding is that this is essentially a "daily composite" image, summing up the total ash AOD detected from the set of IASI ash-detections during the 24-hour period 00:00UT to 23:59UT on 10th April 2021. Assuming that's correct, please change the start of the caption from "Ash optical depth" instead to "Daily composite images of ash optical depth".**

For clarity, especially given the changes made to the post processing described above, we have added a paragraph to section 2.1.2 (before the describing the gridding process) to explain how the results are divided:

*'The results have been divided into descending (satellite travelling N to S; ~9:30 am local overpass time at the equator) and ascending (satellite travelling S to N; ~9:30 pm local overpass time at the equator). This approach can lead to artefacts in the results at the point where the data crosses into the next day. This is particularly notable in the ascending orbits at La Soufrière, with this date change occurring around the location of the volcano. To minimise this impact the descending and ascending results have been offset from each other: the descending results are a 24 hour composite of the descending nodes of orbits starting on each date, while the ascending results are compiled from 24 hours of ascending nodes of orbits starting from midday on each date and up until midday of the following day. In this way the artefact is moved to the other side of the globe so minimising its impact. In this paper, the ascending results are referred to by the start date.'*

We feel this should give an adequate explanation of how the results have been divided but we've also add the following to the figure 1 caption:

*'The results are a composite of multiple orbits which have been divided into descending (~ 9:30 am local overpass time at the equator) and ascending (~9:30 pm local overpass time at the equator). The ascending results are a composite of orbits from 12:00 UTC on the 10 April to 12:00 UTC on 11 April. For further details see the main text.'*

We have added more detail to many of the figure/table captions to help the reader better understand the difference between the ascending and descending data.

**31) Section 2.1.2, line 119 -- Please clarify the time-interval for the two images in Figure 1. My understanding from the text is that both are sampling differently from within the same time-interval, with this presumably being the 24-hour period from 00:00UT to 23:59UT on 10th April 2021? Assuming that's the case, then please change "an example of the" instead to "an example daily composite image of the", and add ", 12 to 36 hours after the first explosive eruption at 12:41 UT on the previous day."**

As outlined in our response to the previous comment we have now added a description of how the results were divided into ascending and descending earlier in the section.

We have not added 12 and 36 hours as suggested as the time is not consistent for all orbits.

**32) Section 2.1.2, line 119-121 -- The two sentences beginning "The retrievals show" and "It is therefore likely" are communicating a related point, and this is best explained simply within 1 sentence, also being clear this follows from the analysis from Carboni et al. (2012) explained in the preceding paragraph. The 3rd sentence of this paragraph is essentially repeating the information given in the caption to Table 1, and then that sentence can be deleted, simply citing the Table 1 within the re-worded merged sentence of the currently first and 2nd sentences. Suggest then to combine the first two sentences of the para, re-wording from "volcano. It is therefore likely..." instead to "volcano, analysis of the percentage ash-detections (Table 1) showing the 10th April retrieved SO2 column amounts must be affected by ash."**

Based on this comment and comment 33, we have rewritten this paragraph to try and make it more concise and to avoid repeating information in table 1:

*'Figure 1 shows examples of an optically thick ash plume travelling east from the volcano on 10 April 2021. Analysis of the percentage ash-detections (table 1), show that the $SO_2$ column amounts may be strongly affected by the presence of ash, and so the column amount and mass values presented in this paper should be considered minimum estimates, especially in the first few days after the eruption began (before the ash falls out).'*

The sentence about spectral similarities with desert dust has been moved to the paragraph before this.

**33) Section 2.1.2, lines 125-126 -- Re-word "gives some information" to better communicate the basis of this. Suggest to change this to "gives a strong quantitative indication of", and then change "impact of ash" to "substantial impact of ash" and "retrieved results" instead to "retrieved SO2 burden".**

See comment 32.

**34) Section 2.1.3, lines 130-136 -- The use of the symbol lamda here for e-folding time is confusing. For first-order loss rate equations, the term lambda should be used for the loss rate (i.e. the term with units of "per unit time") -- a lambda symbol would be better, the greek letter l being short-hand for "loss-rate". For the e-folding timescale, the tau symbol would be much more appropriate, the greek letter t then being shorthand for "timescale". Please change all instances of lamda in this section of text instead to tau.**

Done

**35) Section 2.1.3, line 133 -- Insert "average" between "of the" and "e-folding time" and insert "over a given period" at the end of the sentence.**

Done

**36) Section 2.1.3, lines 133-134 -- It's not correct to say that approach "neglects the variable flux of SO2 emitted from the volcano". The approach derives an average over the period there has been that variable flux of SO2 emitted from the volcano. It's better to make a positive point here, that the optimal estimation scheme enables to derive time-varying information. Suggest then to re-word the current" this approach neglects the variable flux of SO2 emitted from the volcano" instead to "for an eruption with multiple significant SO2 emissions episodes, a time-varying SO2 emission flux can be derived."**

We've rewritten this:

*'For an eruption like La Soufrière, where there are multiple emission events, the total SO₂ mass is a function of both the average e-folding time and variable SO₂ flux (f):*

$$m_i = m_{i-1} e^{-\frac{1}{\tau}\Delta t} + f\tau (1 - e^{-\frac{1}{\tau}\Delta t})$$

*Where i is the time step and Δt is the time interval between measurements.*

*Carboni et al. (2019) uses Eq. 2 within an optimal estimation approach to estimate both the SO₂ flux at each time step and an average e-folding time for the entire period. This approach has been applied here to the total masses obtained for La Soufrière...'*

**37) Section 2.1.3, lines 134-135 -- The current wording "considers that the total SO2 value is effected by both the e-folding time and variable SO2 flux" needs to be improved, to better communicate the processes involved here. I think the text is indicating that SO2 oxidation is acting to reduce the SO2, whereas the emission is acting to increase the SO2 -- i.e. there are both localised source terms from the emission, and also a first-order loss term (i.e. sink tends broadly to scale with the concentration in the air, with also the concentration of the oxidant).And then in that sense the spelling "effected" is correct (rather than affected).But when the authors say "e-folding time" I think they mean the loss due to oxidation of the SO2, right? In which case, please replace "e-folding time" with "oxidation sink".**

To clarify this we've added a description of what is meant by SO₂ e-folding time:

*'The e-folding time refers to the lifetime of SO₂ in the atmosphere and incorporates the loss of SO₂ due to oxidation and deposition, and where the SO₂ amount falls below the detection limit of the instrument being used.'*

The terms e-folding time, and SO₂ lifetime, have been widely used in the literature and we are reluctant to deviate from this.

**Similarly, the wording is better to state "the multiple SO2 emissions from large-magnitude explosions in the 14-day period" or similar. The word "value" there I think means the total SO2 burden, right? Or do they mean the emissions flux at each explosion? Assuming it's the former, suggest then to re-word to "is effected by both the oxidation sink and the multiple SO2 emissions from large-magnitude explosions in the 14-day period". Or something more specific such as this.**

We've edited this section and this now reads:

*'For an eruption like La Soufrière, where there are multiple emission events, the total SO₂ mass loading is a function of both the average e-folding time and variable SO₂ flux (f) ... '*

With an explanation of the e-folding time given above.

**38) Section 2.1.3, lines 139 -- Insert "emissions" before "flux" (after "estimates of the"), and insert "for the entire period" at the end of the sentence.**

We have added 'SO₂' rather 'emissions' and edited this sentence to fit with other changes:

*'Carboni et al. (2019) uses Eq. 2 within an optimal estimation approach to estimate both the $SO_2$ flux at each time step and an average e-folding time for the entire eruptive period. This approach has been applied here to the total masses obtained for La Soufrière.'*

**39) Section 2.1.3, lines 139-140 -- The two sentences beginning "It was noted..." and "Subsequently, an independent..." should be merged (again they are making related points about the same issue, and then explaining in 1 sentence is better). The wording of the start of the 1st sentence should be improved, and I'm suggesting to make this more general from the current "It was noted that in thie case the.." instead to "In some cases, the...", then changing "was strongly influenced by" instead to the general tense "can be strongly influenced by". And then have the wording of the continuation (former 2nd sentence), re-wording from "the a priori value. Subsequently, an independent estimate..." instead to "the a priori value, and for this case an independent estimate...".**

This has been changed to:

*'The average e-folding estimate can be strongly influenced by the a priori value, and so for this study, an independent estimate of the'*

**40) Section 2.1.3, line 143 -- Re-word "approach to getting flux and e-folding time for the 9 to 22 April 2021" instead to "approach, to generate a time-varying emissions flux for the 14-day emission period (9th to 22nd April 2021)."**

Done

**41) Section 2.1.3, line 144 -- Insert "average" before "e-folding time of".**

Done

**42) Section 2.1.3, line 145 -- Delete "The event start and end times obtained from the ABI data were used to inform this." That information is already communicated in the "as identified with the ABI data" in the preceding sentence.**

Done

**43) Section 2.1.3, lines 148-151 -- Yet again, there are 3 short sentences here that relate to the same issue, again better communicated in one carefully worded long-ish sentence. In fact the 3rd of the sentences here is simply noting the way the lower-bound estimate was derived, and this can be deleted, having the re-worded sentence refer both upper-bound and lower-bound estimates.**

We have turned this into two sentences (see below) but feel that one sentence would lead to loss of clarity.

**The term quadrature will be unfamiliar to some readers, and the first required edit here is to change "summed in quadrature and then multiplied by delta-t" instead to "summed in quadrature (i.e. a sum multiplying each event-flux by delta-t)".**

We've changed this to:

*'... the individual flux errors are summed in quadrature (i.e. a quadratic mean of the errors), and then multiplied by Δt.'*

Multiplying by delta is a second step which converts the flux from Tg per day to Tg per half day.

**And then the continuation to the follow-on sentence re-worded from "delta-t. An alternative estimates of the maximum total erupted mass value" instead to "delta-t), with upper- and lower-bound total emissions estimates obtained by summing the corresponding +/- standard deviation individual SO2 emissions at each time (see Figure 2)." or similar.**

While related, combining both methods of estimating the errors into one sentence loses clarity. However, we have rephrased the second method as:

*'An upper and lower bound of the total emission estimate is obtained by summing the SO$_2$ flux plus/minus the errors at each time step (excluding negative values), and multiplying by Δt'*

**44) Section 2.2.1, line 155 -- insert comma before "launched in 2016".**

Done

**45) Section 2.2.1, lines 155-158 -- These 2 sentences can easily be worded into 1 sentence, which much better communicates the issue here (rather than 2 compartmentalised short-sentences with two parts of the same issue). This is easily done by changing "the Caribbean. In addition, it covers the.." instead to "the Caribbean, extending across the...". Change also "of the Atlantic Ocean" to the more specific "of the North Atlantic ocean".**

Done

**46) Section 2.2.1, lines 159-160 -- Change "Seven channels between 7.3 and 13.3 microns mean that the instrument.." instead to "Having seven channels between 7.3 and 13.3 microns means that the instrument..."**

Done

**47) Section 2.2.1, lines 164-165 -- This short sentence here refers to the GOES team enacting a higher-temporal resolution "mesoscale" data-stream focused in the region around the La Soufiere volcano. Presumably this decision was made by a steering group for the GOES operating procedures, in response to the substantial eruption having taken place. This would be a good opportunity to cite some further information on the basis of this. Is it possible to cite a report or paper that explains some background to there being this flexibility to enact localised high temporal-resolution data-flows as a response to the eruption? Or is there a webpage on**

**the GOES/ABI website that explains this? A slight extension to this sentence would enable readers to appreciate the forward-planning discussions that enabled this to happen.**

We've added a little more information to this paragraph which gives a more general introduction to the movement of these mesoscale regions to respond to events:

*'In addition, on GOES-16 there are two moveable mesoscale regions, covering an area of 1000 x 1000 km (at subsatellite point), which can provide data every minute (Schmit et al. 2017). These are moved to provide higher temporal coverage for events such as severe weather, hurricanes and forest fires (Schmit et al., 2017).'*

**48) Section 2.2.1, line 166 -- change "event" to "events".**

Done

**49) Section 2.2.1, line 168 -- re-word "have been used for the identification of the start and end times of the eruptive events" to a more succinct wording such as "have been used to identify the start and end times...."**

Done

**50) Section 2.2.1, line 170 -- insert comma before "it has not been".**

Done

**51) Section 2.2.1, line 170 -- change "Instead it has been used to..." instead to "Instead, the ABI data has been used to..."**

Done

**52) Section 2.2.2, lines 173-174 -- This sentence needs to be re-worded to make clear to the reader whether these RGB images are generated operationally within the GOES team, or if this is something specific the authors have done for this analysis. My understanding is that these ash RGB images are already available to view, based on an established methodology from the peer-reviewed literature. Please improve these initial sentences to refer to the published studies that have established this methodology, and mention whether the images are already supplied from the GOES team (and the extent of the analysis carried out by the manuscript authorship team).**

We've added a line in section 2.2.1:

*'Recent images produced with this instrument can be accessed at https://www.star.nesdis.noaa.gov/goes/fulldisk.php?sat=G16 (webpage last accessed on 05/04/2023)'*

And then in section 2.2.2:

*'Utilising the Satpy Python package, true and false colour maps have been produced from the ABI data for the eruption period (9-22 April 2021). The false colour images have been constructed by assigning the 12.3 - 10.3 µm, 11.2 - 8.4 µm and 10.3 µm channels to red, green and blue respectively (for further information see GOES-R, 2018).'*

A reference to the GOES-R quick guide is also included in the caption of table 2.

**53) Section 2.2.2, line 188 -- "inference of the character of the event" needs to be clearer what is meant by "character". I think you mean the information included in the "Notes" column of Table 2, and briefly note (in brackets after the "character") what you mean here.**

On reflection we think this sentence is a little confusing. We had meant that we can identify similarities between the different events based on the duration, type of plume produced, which is how we were able to divide the eruption into four phases. Given this is covered in the results section we've decided to remove this sentence.

**54) Section 2.2.2, line 191 -- Change "It is also possible that lower level eruptive activity may not be identifiable" to be clearer here. Firstly, by "lower level eruptive activity" I think you mean explosive events of lesser magnitude -- i.e. "lower-magnitude explosive events" or so. Secondly, you've writen "possible" but I think "likely" seems more consistent with what is meant here? Please re-word accordingly.**

Changed to:

*'It is also likely that lower-magnitude explosive events or degassing between events may not be identifiable...'*

**55) Section 2.2.2, line 192 -- insert commas after "Note that" and "full disc" to improve the grammar of the sentence on this line.**

Done

**56) Section 2.2.2, lines 193-194 -- Re-word "the rough measurement start and end times". The subsequent wording indicates the method here is quite precise, so I'd suggest simply to delete the word "rough". Similarly delete "roughly", as again this seems relatively precise. I realise the aim here is to communicate that, with the frequency of the data within 10-minute intervals, the calculation is more approximate than during the minute time-resolution "mesoscale" data-flow. But that can be communicated in the follow-on sentence (see 57 below).**

We've deleted the word 'rough' and changed 'roughly' to 'approximately' as the estimate of the time over La Soufrière is not exact:

*'Note that, for the full disc, the measurement time over La Soufrière has been computed based on the latitude of the volcano and the measurement start and end times, approximately 243 seconds from the start time.'*

**The follow-on sentence beginning "This is estimated to be" makes for very poor sentence construction, and again merging the 2nd very short sentence to slightly extend the first sentence would seem to improve the readability of the text. Suggest then to change "end times. This is estimated to be roughly 243 seconds (4 minutes and 3 seconds) from the start time" instead to "end times, roughly 243 seconds from the start time". It is not necessary to give the minutes and seconds  translation of this, the SI unit for time in seconds, and all readers will be familiar with converting seconds to minutes.**

Done

**57) Section 2.2.2, line 194 -- insert "approximate" before "times for..".**

Done

**58) Section 2.2.2, line 195 -- change "where the measurement start time has been used" instead to "where the 1-minute time resolution ensures the timing is highly accurate." or something like this.**

This has been rewritten as:

*'No such adjustment has been made to the mesoscale results, where the 1-minute temporal resolution ensures a higher accuracy.'*

**59) Section 2.2.2, line 198 -- Improve the wording "This method will now be referred to as the..." instead to "We refer to this as the...".**

In the manuscript we have not used first person so for consistency we have rewritten this as:

*'Here this is referred to as the "Brightness Temperature (BT) method"*

**60) Section 2.2.2, lines 202-203 -- "The box size should remove any effect due to parallax". Re-word to state the size of the box, and insert "significant" between "any" and "error".**

Done. This now reads:

*'The 0.1° box size should remove any significant effect due to parallax'*

**61) Section 2.2.2, lines 205 to 217 -- There needs to be citations given for most of these caveats, where this effect is explained in further detail. Also, re: the minimum BT method -- please can the authors clarify whether this method still works for large magnitude explosive events that penetrate deep into the stratosphere? I am not so familiar with the literature here, but I note the discussion of this issue within Woods and Self (1992), in relation specifically to very large magnitude eruptions penetrating the stratosphere.**

Re-reading this section, we opted to present these limitations in a table which allowed the inclusion of references without significantly disrupting the flow of the text. It also helps to better communicate mitigating measures taken. Woods and Self (1992) discuss a case where the lower BT temperatures (caused due to overshoot) obtained with the satellite data do not intersect with the temperature profile. This would fall under 'method assumes that the ash is at an equal temperature to the surrounding air'. We have added this as an example.

**Table 3.** Summary of the sources of error associated with estimating the height of volcanic plumes by comparing the brightness temperature at 11.2 $\mu$m (ABI channel 14) with a temperature profile. In this paper this is termed the 'Brightness Temperature (BT) method". These limitations are discussed in several papers (Oppenheimer, 1998; Prata and Grant, 2001; Zakšek et al., 2013).

| Limitation | Explanation | Mitigation (if applied) |
|---|---|---|
| Single height estimate | In this application this method only returns a single height rather than reflecting the heights across the plume. In addition, the minimum BT has been selected for each eruptive event and so the reported values do not reflect the variation in height during the eruptive event. | |
| Temperature profile | A good quality temperature profile, close to the volcano, is required (Oppenheimer, 1998; Zakšek et al., 2013) | An ECMWF ERA5 temperature profile is interpolated to the volcano's location. |
| Optically thin ash | When the plume is optically thin, upwelling radiation from beneath or within the plume contributes to the measured brightness temperature (Glaze et al., 1988; Zakšek et al., 2013) | This effect is minimised here by selecting the pixel with the coldest brightness temperature. |
| Poor correspondence between temperature profile and satellite measurement | The method assumes that the ash is at an equal temperature to the surrounding air, if this is not the case, then the method will not perform well. Oppenheimer (1998) notes that the plume may not have reached its maximum height or that the momentum of the plume means it may have overshot the thermally equilibrium level. Some eruptions (e.g. El Chichón and Mt. St. Helens ) the ash plume is much colder than the surrounding air and so the heights obtained were incorrect (Woods and Self, 1992; Holasek et al., 1996a). | If there is no intersection with the temperature profile no result is reported. |
| Multiple solutions | Multiple solutions can arise due to multiple intersections with the temperature profiles (e.g. above and below the tropopause, and other temperature inversions), in such cases, additional data sources are required to clarify the result (Oppenheimer, 1998) | In this study multiple solutions are reported. Further analysis using the plume's direction of travel and the wind profiles have been used to try and determine which solution is more appropriate (see main text). |
| Isothermal atmospheres | The method is limited in parts of the atmosphere with little temperature variation (Holasek and Self, 1995). A similar effect is observed using other infrared retrieval techniques (e.g. Prata et al., 2022). | |
| Above cloud absorption | This method does not account for any gaseous absorption above the cloud top | This is included within the 1 K uncertainty |
| Meteorological cloud | Where there is meteorological cloud overlying the plume, heights will reflect the height of the meteorological cloud layer. | A single height is reported for each event, typically near the start of the event when the ash is not obscured by cloud. |

**62) Section 2.2.2, line 230 -- re-word "compared against the plume vector components to assign a degree of confidence". Firstly, I don't know what is meant here by "plume vector components". You mean the horizontal velocity components from the motion of the plume? Please clarify what is meant here. Also, re-word instead to "compared to the plume vector components (i.e. horizontal plume motion from successive geostationary images), and a degree of confidence assigned accordingly". Please clarify re: the plume vector components, and make amendments to the suggested re-wording accordingly.**

You are correct that the plume vector components here refer to the horizontal velocity components. As suggested, this has been rephrased to improve clarity:

*'The wind directions at the heights obtained with both BT methods are compared against the horizontal velocity components from the motion of the plume (based on distance travelled in 1 hour or 30 minutes, as above) to assign a degree of confidence to the height results for each eruptive event'*

**63) Section 2.2.2 lines 233-234 -- Improve this very short final sentence of this paragraph -- re-word to "the different sampling periods assessing different portions of the plume" or similar.**

We have rewritten this and it is now as a separate paragraph and refers to the Sparks et al. (in press) paper as well:

*'There are differences between the heights obtained with the "BT method" applied in this study and the results obtained using the BT method found in Horvath et al. (2022), which may arise from different sampling approaches. This is also the case for the BT heights presented in Sparks et al. (in press) where differences may also arise due to the different meteorological data and channels used.'*

**64) Section 3.1, lines 254-255 -- The sentences beginning "This is seen..." and "This corresponds well..." should be re-worded to 1 sentence, and this can be done easily, changing "9 April. This corresponds" instead to "9 April, and corresponds well", the resulting merged sentence being easier to read.**

Done

**Correct typo also with the bracket in the wrong place in the current wording "with the 08:41 (LT 12:41 UTC)..." --> correct this to "with the 08:41 LT (12:41 UTC)..."**

Done

**65) Section 3.1, lines 255-256 -- Correct "Relatively short lived lasting..." to "Relatively short-lived, lasting..."**

Done

**66) Section 3.1, line 260 -- Correct "second lower altitude" instead to "second lower-altitude", and add comma before "travelling to the west".**

In response to a comment by Referee #1 we expanded the sentence explaining the reasons for difference between ground based and satellite height estimates. To do this we moved the sentence about the lower altitude plume earlier in the paragraph:

*'During this time two plumes are evident: a low altitude plume can be seen travelling to the west from the volcano, while the main plume…'*

**67) Section 3.1, line 263-264 -- Correct "the plume direction and the wind profile does not help" to "the plume direction and wind profile do not help".**

Done

**68) Section 3.1, line 279 -- Insert "distinct" after "further seven".**

Done

**69) Section 3.2, lines 294 to 308 -- It is thie section of the text that requires improvement to refer to the QBO (see general points above).**

**In particular the text on lines 295 to 298 requires improvement, as it does not give any scientific interpretation of the initial plumes being transported entirely eastwards (westerly flow) whereas the later daily composite images for 13th to 17th April indicate a portion of the plume is being transported westward. There needs to be a better link between the wind direction profile Figure 4a and what is seen here in the Figure 7 maps. Specifically, whether the westward transport is indicating transport at a different level than the initial plumes on the 9th to 11th? Figure 8 shows the plume heights for the 10th to 11th April, and the altitudes for the later eruptions are given in Table 3 with the tropospheric and stratospheric "solutions". But the text needs to be clearer what this means in relation to the dispersion of the column seen in the Figure 7 maps. Is the mapping indicative of the upper tropospheric portion of plumes is transported in a different direction to the stratospheric plumes, as suggested by the Figure 4a? Mention of the QBO and the suddent shift in wind direction into the lower stratosphere should be mentioned here.**

Following these comments, we have made a few additions to different sections of the manuscript.

First, we've added a line to section 2.2.2 which mentions the QBO:

*'The easterly winds in the stratosphere are a characteristic associated with a phase of the Quasi-Biennial Oscillation (QBO): alternating strong easterly and westerly zonal winds around the equator which propagate through the stratosphere to the tropopause (Reed et al. 1961, Baldwin et al, 2001).'*

We've added some lines to the second paragraph of section 3.2 which relates the dispersion of $SO_2$ to the wind direction, and how this might relate to the plume height:

*'... A stronger signal is then visible in the descending orbits on 10 April fanning out to the east of the volcano across the North Atlantic. The general east and south-eastward transport of the plume between 9 and 11 April implies that the bulk of $SO_2$ has been emitted into the troposphere, with Fig. 4a indicating wind directions between ~90-140° dominating in the troposphere between 8 and 17 km... On the 12 April, while the bulk of the plume is still advancing towards the east (Fig. 7), a fraction of the plume travels to the south and west of the volcano. The wind directions shown in Fig 4a, imply that for westward transport, either some $SO_2$ has been emitted into the lower parts of the troposphere (less than 5 km) or more likely that some $SO_2$ has been emitted, or has been lofted, into the stratosphere.'*

We've added an additional line at the end of the paragraph to indicate that there is an eruption at Sangay in Ecuador which produces a plume that cannot be easily distinguished from the plume from the La Soufrière plume. This perhaps slightly exaggerates the westward movement.

*'Note that there is an emission of $SO_2$ from an eruption at Sangay in Ecuador from 12 April (GVP, 2021d). This combines with the plume from La Soufrière and the two cannot be easily distinguished from each other from 13 April.'*

Following this comment, we questioned whether we would expect to see a greater variation in heights in the upper portion of the plume given the wind direction variability and as is seen in Koukouli *et al.* (2022). We reran the retrieval with a range of different setups (varying the first guess height and the thicknesses) to see if this altered our results. Very little change was seen in the results and so no change has been made

to the manuscript. However, in section 3.4 we've added further detail relating the plume transport direction to the height information:

*'There is no obvious gradient to the heights in the upper part of the plume as can be seen in IASI SO$_2$ height measurements shown in the supplementary material of Koukouli et al. (2022) for 10 and 11 April (using the Clarisse et al., 2014 method). Koukouli et al. (2022) show heights increasing from south to north of the plume: which matches well with the wind directions shown in Fig. 4a. The IASI retrieval used here relies on temperature and water variations in the atmosphere, which do not vary as significantly around the tropopause, which may affect the results. Multiple retrieval setups were explored (including varying the retrieval first guess height and varying the plume thickness) but the results were similar in each case. Nevertheless, there is a broad agreement with Koukouli et al. (2022) which reports average heights of 15.7 ± 1.16 km for IASI using the Clarisse et al. (2014) retrieval and 14.94 ± 3.87 km from TROPOMI (Hedelt et al. 2019 method) (note that these averages are based on a subset of the plumes). Additionally, there is agreement with the 13 to 15 km injection heights obtained by Esse et al. (2023) ...'*

**70) Section 3.3, lines 310-311 -- Again, these first 2 sentences of the paragraph are both very short, and I do not understand why the text has been structured in this way. Please change "Fig. 9c. The mass is computed for the following region..." to "Fig. 9c, computed for the region...", and delete the 2nd "region" at the end of the revised single sentence.**

This has been amended to combine the sentences:

*'A total mass timeseries, derived from the IASI iterative retrieval output for the -45 to 45° N and -180 and 180° E region, is shown in Fig. 9c.'*

**71) Section 3.3, line 311 -- Delete "which" after "in Peru)" and change "at the same time" to "at this time", also changing "and whose plumes entered this box are also be.." to "and any SO2 from these eruptions entering the region will also be..".**

This now reads:

*'Note that other volcanoes (e.g. Sabancaya in Peru; Sangay in Ecuador) were erupting at this time and any SO$_2$ from these eruptions entering the region will affect the total mass, e-folding and flux estimates'*

**72) Section 3.3, line 313 -- change "small plumes" to "small plumes, and at lower altitude" (assuming that is the case).**

The height of the plume is not relevant in this case as a large emission, lower in the atmosphere could still lead to an overestimation of the mass. For clarity we have changed this to:

*'However, given these are smaller emissions than La Soufrière, their impact is negligible and within the reported errors.'*

**73) Section 3.3, lines 317-320 -- this segment of text is where there needs to be a better scientific interpretation of the results, in relation to the processes occurring as the multiple explosive-emission SO2 plume disperses. A specific suggestion is to have the sentence "The fact that the total SO2.." start a new paragraph, and expand this sentence to refer to the oxidation of the SO2, and the studies mentioned in the general comments for the studies I mentioned interpreting the progressing SO2 burden within volcanic clouds large-magnitude explosive eruptions (Pinatubo, Kelut, Hunga-Tonga). There should be mention of the potential accelerated SO2 oxidation**

**from reactions on the surface of ash particles (heterogeneous chemistry), referring to the Zhu et al. (2020) study.**

In section 3.3, we've added a sentence to indicate that the $SO_2$ amount is decreasing over time. We've then moved the discussion of the e-folding time from section 3.4 to section 3.3. In this we've also outlined the different variables affecting the $SO_2$ lifetime and added a sentence at the end which indicates that the lifetime obtained here may be affected by ash:

*'From 13 April the total mass of $SO_2$ is shown to fall: as the $SO_2$ is removed from the atmosphere through deposition, by conversion to sulfate aerosol, or dilution below the detection limit of the instrument. The e-folding time used here (see section 2.1.3) describes this loss process. This varies with a number of factors including the latitude of the volcano, the injection height of the plume, meteorological conditions, cloud cover, water vapour, season and the presence of ash (Carn et al. 2016, Zhu et al. 2020, Schmidt et al. 2022, Zhu et al. 2022). Typically, the e-folding time varies from hours to days in the lower troposphere to weeks in the stratosphere. A first estimate of 5.47 days for the e-folding time was estimated by fitting Eq. 1 to the IASI total $SO_2$ masses between 23 to 30 April 2021. Following the application of the Carboni et al. (2019) method to compute the flux and average e-folding time, the average e-folding time was adjusted to 6.07±4.74 days. This is in line with other eruptions including Jebel at Tair (2007) and Merapi (2010), both volcanoes in the tropics and which emitted plumes between 15 and 18 km, and had e-folding times of between 2 and 4 days (Carn et al., 2016). Given that there was ash emission during the La Soufrière eruption, it is possible that the e-folding time could have been reduced as a result of accelerated oxidation of $SO_2$ due to reactions on the ash surface, as was seen for the Kelut eruption in a study by Zhu et al. (2020). Some of the $SO_2$ emitted during the La Soufrière eruption was converted to sulfate aerosol, as is shown in Babu et al. (2022) and Bruckert et al. (2023).'*

We do not feel that comparisons with other eruptions, such as Pinatubo or Hunga Tonga-Hunga Ha'apai which have very different characteristics, are relevant to the science discussion in this manuscript (which focuses on the similarities with the 1979 La Soufrière eruption) and would bring the discussion significantly out of scope of the present study.

**74) Section 4, line 415 -- Improve the sentence beginning "This is similar to this study which reports..." -- probably this can be joined with the preceding sentence for a more coherent and readable statement.**

This has been changed to:

*'As mentioned above, Shepard et al. (1979) gives height estimates for individual eruptive events ranging between 8 and 18.7 km, similar to the heights reported here for the April 2021 eruption.'*

**75) Section 5, line 431 -- Re-word "emitted large plumes of ash and SO2 into the atmosphere" to better state the science narrative of the study re: the study generating an eruption chronology across the several large magnitude explosive events in the initial days, through to a full 14-day dataset, comprising 4 distinct eruption phases. A specific suggestion would be to expand "emitted large plumes" instead to "comprised several large-magnitude explosive events, each generating tropopause-penetrating plumes of volcanic SO2 and ash", with later phases continuing to emit SO2 into the upper troposphere up to xx days after the initial explosion". Or similar wording to this.**

We've changed the opening sentence to:

[revised manuscript text omitted]

---

## Referee Report (RR1)

**Review of "A satellite chronology of plumes from the April 2021 eruption of La Soufrière, St Vincent"**

The paper is interesting and offers a comprehensive picture of the eruption of La Soufriere of April 2021 regarding the volcanic clouds seen from satellite (IASI and ABI sensors). I have just some questions and some comments regarding the plume height retrieval from ABI images. It is in my opinion suitable for publication after minor revisions.

General comment about the position of figures and tables: it happens quite frequently within the manuscript that the figures and tables are positioned quite far from the text where they are explained and commented. Please, if possible, move them to improve readability. For example:

- Table 1 is positioned between lines 119-120, while it is explained at line 142
- Figure 2 is positioned in 2.2.1 section while it is described in 2.1.3 section
- Figure 3 is positioned in 2.2.1 section while it is described in 2.2.2 section
- Etc.

- Line 34: I think that coordinates are usually written with latitude first, followed by longitude. The same at line 361 for Sabancaya.

- Line 86-87: "three absorption features v1, v3 and v1+v3, centred at 8.7, 7.3 and 4 μm respectively". Please add a reference also for this.

- Line 209: instead of "Careful examination", I would prefer "Careful visual inspection" to enhance the presence of an operator looking at the maps

- Lines 215-126: "In general, the end time was determined when the plume moved away from the volcano". Please clarify this aspect: the end time was determined when there was almost one "clean" pixel between the volcano and the plume?

- Lines 219-220: I would rephrase the sentence in "the measurement time over La Soufrière has been precisely computed based on the latitude of the volcano adding 243 seconds to the sensing start time". This does not seem an "approximate" but a very precise correction. Anyway, there is a little discrepancy with the sentence below: "No such adjustment has been made to the mesoscale results, where the 1-minute temporal resolution ensures a higher accuracy". I totally agree with this (1 minute is a good accuracy) but then why you correct the fulldisc with 1 second of precision? A simpler 4-minute correction for fulldisc would be more consistent with the "no correction" applied to mesoscale images (producing a 1-minute precision time for all images).

- Line 224: please make explicit how many pixels correspond to the 0.1 deg box. 5x5 pixels around the volcano?

- Line 224: about the ERA-5 profiles: which is the temporal step you considered? 1 hour?

- Line 244 and fig. 4a and 4b: please clarify how you have computed these averages and standard deviations. Do you considered all the ERA-5 profiles between 9 and 22 April, 1 hour step, interpolated at the volcano position? Why you computed the standard deviations for wind direction and not for wind speed too? In fig.4a, I expected that, for each height level, the red lines (+/- standard deviations) would be equally spaced with respect to the averages (black lines). Why are not? The caption of fig. 4b is wrong.

- Line 251-252: the visually identification of the centre of the cloud seems a bit arbitrary and a not very accurate method. For future works, I suggest using something more objective, like the coldest pixel of the cloud or the centroid around the minimum, etc. Anyway, how do you choose between 1 hour or 30 minutes time ranges? Why do you choose these values and not, for example, 2 or 3 hours? Please give a justification for this choice.

- Fig.5: In my opinion, these figures are not so clear. As titles, instead of using "Eruption 1", "Eruption 2" and "Eruption3", I suggest putting the date and time of the ERA-5 profile considered. The use of both BT solutions ("optically thick" and "plume centre") is not so relevant here and can create confusion. I suggest putting only the BT optically thick solutions.

- Table 3: "If there is no intersection with the temperature profile no result is reported." I am not sure that this aspect can be considered a "mitigation" when the plume is not thermally balanced with the surrounding air. It may happen that the plume temperature is higher or lower than air temperature but still have a point of intersection with the profile.

- Table 3: "A single height is reported for each **event**, typically near the start of the **event** when the ash is not obscured by cloud." Use "eruption" in place of "event" to avoid repetition.

- Table 5: I suggest modifying this table, which is not very significant at the moment:

> 1) What is the valuable contribution of considering the height deriving from the BT plume centre too? The heights values are generally very similar (a bit lower for troposphere, a bit greater for stratosphere) of that obtained from BT optically thick. The reason could be that after 1 hour, the plume is probably less opaque and the temperature is higher. Therefore, in my opinion the plume centre height solutions are not necessary.

> 2) The errors reported in table 5 are obtained considering only +/- 1 K, I suppose. This uncertainty is used to consider the instrument noise and gaseous absorption above the cloud (as you said at line 235). But, as you have optimally described in table 3, many others sources of error for BT method are possible. Errors of 100-200 m seem underestimated for this method. I suggest using at least +/- 2K or +/- 3K to give an estimate of the overall errors or, alternatively, clearly explain that the reported values do not take into account all other possible sources of error for this method.

> 3) In table 5 I suggest highlighting (with bold character for example) the more confident height value (tropospheric or stratospheric, based on the wind comparison), to help the reader in identifying the most probable height value more easily.